# Falsification of Unconfoundedness by Testing Independence of Causal Mechanisms

**Rickard K.A. Karlsson** [1]   **Jesse H. Krijthe** [1]

## Abstract

A major challenge in estimating treatment effects in observational studies is the reliance on untestable conditions such as the assumption of no unmeasured confounding. In this work, we propose an algorithm that can falsify the assumption of no unmeasured confounding in a setting with observational data from multiple heterogeneous sources, which we refer to as environments. Our proposed falsification strategy leverages a key observation that unmeasured confounding can cause observed causal mechanisms to appear dependent. Building on this observation, we develop a novel two-stage procedure that detects these dependencies with high statistical power while controlling false positives. The algorithm does not require access to randomized data and, in contrast to other falsification approaches, functions even under transportability violations when the environment has a direct effect on the outcome of interest. To showcase the practical relevance of our approach, we show that our method is able to efficiently detect confounding on both simulated and semi-synthetic data.

## 1. Introduction

Using observational studies to estimate treatment effects is a ubiquitous yet challenging task in many disciplines, such as medicine (Hernán & Robins, 2006) or social sciences (Athey & Imbens, 2017). Whereas there exists a rich literature of methods for treatment effect estimation in the observational setting (Bang & Robins, 2005; Wager & Athey, 2018; Chernozhukov et al., 2018), all methods have in common that before a causal effect can be estimated, often untestable conditions need to hold. One such condi-

tion is that we assume there is *no unmeasured confounding*, meaning that there are no unobserved factors that have both an influence on the treatment and on the outcome of interest that are not accounted for by the method. If unmeasured confounders are present, our treatment effect estimates are likely to be biased and inconsistent (Greenland et al., 1999). This can have serious downstream consequences such as unknowingly recommending a non-effective or, even worse, potentially harmful treatment policy. Unfortunately, without making further assumptions, it is in general impossible to verify all assumptions needed to identify treatment effects from observational data.

In this work, we investigate a novel strategy for falsifying unconfoundedness. Specifically, we focus on the common scenario where observational datasets are collected from different heterogeneous sources, which we refer to as *environments*. Each environment corresponds to distinct study populations, due to factors such as geographical differences that results in distribution shifts between the environments. We propose a falsification strategy based on the assumption that these distribution shifts stem from independent changes in the underlying causal mechanisms. This idea is grounded in the principle of independent causal mechanisms (ICM) (Janzing et al., 2012; Peters et al., 2017), which posits that a causal system comprises autonomous modules that do not inform or influence each other. Assuming independent causal mechanisms has been leveraged to, for instance, improve causal structure learning (Huang et al., 2020; Guo et al., 2024a) and understand model behavior in statistical machine learning (Schölkopf et al., 2012). However, the implications of assuming independent causal mechanisms to treatment effect estimation problems has received far less attention.

Our proposed falsification strategy leverages a key observation that unmeasured confounding can cause observed mechanisms to appear dependent (Janzing & Schölkopf, 2018; Karlsson & Krijthe, 2023; Mameche et al., 2024b; Reddy & Balasubramanian, 2024). If we assume that the underlying causal mechanisms should be independent, contrary to what is observed, it follows that any apparent dependencies could be the result of unmeasured confounding. This observation motivates the central research question of this paper: *How*

[1]Department of Intelligent Systems, Delft University of Technology, the Netherlands. Correspondence to: Rickard Karlsson <r.k.a.karlsson@tudelft.nl>.

*Proceedings of the 42nd International Conference on Machine Learning*, Vancouver, Canada. PMLR 267, 2025. Copyright 2025 by the author(s).

*can we efficiently test causal mechanism independencies to falsify the conditions required for treatment effect estimation in settings with multi-environment data?*

**Contributions** By formalizing the problem using a Neyman-Rubin causal model for multi-environment data, we show that falsification of unconfoundedness is possible by testing dependencies between causal mechanisms directly by combining the principle of independent causal mechanisms with functional assumptions on the mechanisms. In this model, we prove that the presence of unmeasured confounding has testable implications in the form of dependencies between the model's observed parameters. Using our theoretical results, we introduce new algorithmic ideas that can be used for falsification: in particular, we propose a two-stage algorithm that statistically tests statistical dependencies between learned model parameters of the treatment assignment and outcome mechanism. We show that our algorithm performs favorably compared to alternative approaches on both simulated and semi-synthetic data.. To showcase the potential applications of our algorithm and clarify what constitutes an "environment", we provide two illustrative examples where we envision our algorithm being used.

**Example 1** In a meta-analysis of multiple observational studies with individual participant data (Riley et al., 2010), our algorithm can jointly test whether an unmeasured confounder is present between the treatment and outcome across all studies. Here, each observational study serves as a distinct environment.

**Example 2** In a single observational analysis involving a multi-level structure in which individuals are nested in clusters and non-randomly assigned to a treatment/control on an individual level, such as students from different schools (Leite et al., 2015) or patients from different hospitals (Goldstein et al., 2002), our algorithm can test whether the conditions necessary to identify treatment effects are violated within each cluster due to unmeasured confounding. In this context, the environment refers to the sub-populations within each cluster of the same observational study.

## 2. Related Works

When discussing the validity of causal assumptions, sensitivity analysis might come to mind. In sensitivity analysis, one hypothesizes departures from the assumption of no unmeasured confounding and investigates how different biases would arise depending on the hypothesized confounder's relationship with treatment and outcome (Cornfield et al., 1959; Tan, 2006; VanderWeele & Ding, 2017). This typically results in bounds on the treatment effect, which is an instance of partial identification (Manski, 2003). However,

while sensitivity analysis probes 'what-if' scenarios regarding potential unmeasured confounding (a process that can always be undertaken), falsification aims to empirically test whether assumptions are violated, based on the observable implications of those assumptions (which is not always feasible). For instance, falsification may involve testing the validity of instrumental variables (Pearl, 1995) or evaluating the compatibility of learned causal structures with observed data (Faller et al., 2024). In this way, sensitivity analysis and falsification are complementary: the former explores possible scenarios, while the latter seeks direct empirical evidence for these scenarios. Despite this, falsification has received comparatively less attention in the literature.

One line of research on falsification in observational causal inference assumes that certain transportability conditions hold, allowing causal effects to be transferred between different environments (Dahabreh et al., 2020b; Hussain et al., 2022; 2023). The basic premise is that, under transportability conditions, comparing treatment effect estimates from multiple observational studies, or from a single observational study and a randomized one, should yield consistent results. If inconsistencies are found, this can be used to falsify the identifiability conditions, assuming the transportability assumptions hold. This idea has been further extended to time-to-event outcomes with censoring (Demirel et al., 2024), as well as for quantifying bias from unmeasured confounding (De Bartolomeis et al., 2024a;b). In contrast, our approach assumes independence of causal mechanisms, which does not require transportable treatment effects or access to randomized data.

Testing for independence of causal mechanisms has been applied in previous work to falsify causal assumptions, such as detecting hidden confounding (Karlsson & Krijthe, 2023) or testing the validity of instrumental variables (Burauel, 2023; Karlsson et al., 2023). Most similar to our work is that of Karlsson & Krijthe (2023), though their method relies on conditional independence testing which is a notoriously difficult statistical problem in itself (Shah & Peters, 2020). To avoid the challenges of conditional independence testing–for instance, losing statistical power as the adjustment set becomes larger–we address this problem by proposing an alternative method that does not rely on conditional independence testing.

Parallel to our ideas on falsification, other approaches have been proposed for detecting or addressing unmeasured confounding, under various assumptions on the setting and data-generating process. For example, when multiple causes are observed (Wang & Blei, 2019; D'Amour, 2019) or when a negative control is available (Lipsitch et al., 2010).

Finally, our work investigates the implications of the principle of independent causal mechanisms, which has a rich literature in causal discovery, particularly in multi-environment

settings (Huang et al., 2020; Perry et al., 2022; Guo et al., 2024a; Mameche et al., 2024a). A closely related line of research assumes the existence of invariant mechanisms across environments (Peters et al., 2016). In contrast, our approach explicitly allows these mechanisms to vary–and, as we will show, such variation is sometimes necessary to enable falsification. Rather than aiming to learn the entire causal structure as typically done in causal discovery, our approach focuses on verifying specific aspects of a partially known structure that is relevant for treatment effect estimation. Recently, Guo et al. (2024b) examined how independent causal mechanisms can lead to identification of certain treatment effects, though they did not address scenarios where causal assumptions are violated, such as in the presence of unmeasured confounders, which we study here.

## 3. Setup

### 3.1. Notation & data structure

For each individual $i = 1, \ldots, n$, we observe baseline covariates $X_i$ in $\mathcal{X} \subseteq \mathbb{R}^d$, a treatment $A_i$ in $\mathcal{A} \subseteq \mathbb{R}$ and outcome $Y_i$ in $\mathcal{Y} \subseteq \mathbb{R}$. We allow the treatment and outcome to be binary or continuous; but to simplify exposition, we will mainly show our results for the continuous case and then discuss how to modify our theory for binary treatments and outcomes when appropriate. We consider observations to be collected from $K$ different environments labeled with $S_i \in \{1, \ldots, K\}$ where $K \geq 2$. We denote $n_s$ as the number of observations from environment $\{S = s\}$ and we define $n = \sum_{s=1}^{K} n_s$ as the total number of observations. Each observation therefore consists of the tuple $O_i = (X_i, S_i, A_i, Y_i)$. Throughout the paper, we will use capitalized letters to denote random variables and small letters to denote their realized values.

We are considering a setting with a composite dataset of observations from separate environments. Each environment represent a different study population where the sampling probability of individual $i$ belonging to environment $\{S = s\}$ can be unknown; this setting is referred to as a non-nested study design (Dahabreh et al., 2020a) . Formally, we consider observations within an environment $\{S = s\}$ to be sampled independently and identically (i.i.d) according to some distribution $(X, A, Y) \sim P(X, A, Y \mid S = s)$. This distribution may vary across the different environments $s \in \{1, \ldots, K\}$. Importantly, observations are not assumed to be i.i.d. if we consider the marginal distribution $P(X, A, Y)$ over all environments. Furthermore, we assume that the environments are related to each other by having a shared, albeit unknown, causal structure, that is: the causal directed acyclic graph (DAG) between the variables $(X, S, A, Y)$ is the same for all $S \in \{1, \ldots, K\}$.

### 3.2. Assumptions for identification of causal effects

To define causal effects of interest, we use potential (counterfactual) outcomes (Rubin, 1974). For an individual $i$, we posit the potential outcome $Y_i^a$ for $a \in \mathcal{A}$ which denotes the outcome under an intervention to set treatment $A_i$ to $a$. For the typical causal analysis in a non-nested study design, the goal is often to estimate the average treatment effect or conditional average treatment effect between two different treatments $a, a' \in \mathcal{A}$ in the underlying population from an environment $\{S = s\}$, that is $\tau_s = \mathbb{E}[Y^a - Y^{a'} \mid S = s]$, resp. $\tau_s(x) = \mathbb{E}[Y^a - Y^{a'} \mid X = x, S = s]$. It is well-known that under certain conditions $\tau_s$ and $\tau_s(x)$ are identified from the observations in environment $\{S = s\}$.

**Assumption 3.1.** We assume the following conditions for each environment $s = 1, \ldots, K$. *Consistency:* if $A_i = a$, then $Y_i^a = Y_i$, for every individual $i$ and every treatment $a \in \mathcal{A}$. *Positivity:* for each treatment $a \in \mathcal{A}$, if $f(x, S = s) \neq 0$, then $\Pr(A = a | X = x, S = s) > 0$. *Unconfoundedness:* for each $a \in \mathcal{A}$, $Y^a \perp\!\!\!\perp A | (X, S = s)$.

Consistency is satisfied when the treatment is clearly defined, ensuring that no hidden treatment variation exist and that there is no interference between individuals. Positivity requires that every possible covariate pattern in the environment $S = s$ has a nonzero probability of receiving each possible treatment option. Unconfoundedness, also referred to as conditional exchangeability, implies there is no unmeasured confounding. That is, the covariates $X$ are sufficient to adjust for in order to identify the causal effect of $A$ on $Y$. In observational studies, assuming unconfoundedness is often considered controversial, requiring strong domain expertise to justify its validity.

When the conditions in Assumption 3.1 are met, both the average treatment effect and the conditional average treatment effect can be identified from the observed data (Hernan & Robins, 2020). Let $\mu_{a,s}(X) = \mathbb{E}[Y \mid X, A = a, S = s]$, then a statistical estimand for the ATE is $\tau_s = \mathbb{E}[\mu_{a,s}(X) - \mu_{a',s}(X) \mid S = s]$ and for the CATE is $\tau_s(X) = \mu_{a,s}(X) - \mu_{a',s}(X)$. Rather than focusing on how to estimate these estimands from data, we will concentrate on how to assess the validity of the conditions that allow us to identify them in the first place. Specifically, in the context of data from multiple environments, we will demonstrate that Assumption 3.1 can be falsified under certain conditions related to distributional shifts across the different environments.

## 4. A novel falsification strategy

### 4.1. Assumptions on environment changes

We consider a general class of models of the treatment and potential outcomes, namely: all linear functions of the

feature representations $\psi(X) : \mathcal{X} \to \mathbb{R}^z$ and $\phi(X, A) : \mathcal{X} \times \mathcal{A} \to \mathbb{R}^{z'}$,

$$A = \alpha_s^\top \psi(X) + \varepsilon_A$$
$$Y^a = \beta_s^\top \phi(X, A = a) + \varepsilon_Y \tag{1}$$

where the noise variables fulfill $\mathbb{E}[\varepsilon_A \mid X, S = s] = 0$ and $\mathbb{E}[\varepsilon_Y \mid X, A, S = s] = 0$. Additionally, under Assumption 3.1, the noise variables are independent $\varepsilon_A \perp\!\!\!\perp \varepsilon_Y \mid S$.

The function class described in (1) encompasses a wide range of complex models, particularly because the representations $\psi(X)$ and $\phi(X, A)$ can involve nonlinear transformations of the variables $(X, A)$. Although we focus on continuous treatment and outcome values to illustrate the core ideas of our falsification strategy, this framework can be extended to generalized linear models that accommodates binary/categorical values. For instance, we can include binary treatment by defining $P(A = 1 \mid X, S = s) = h^{-1}(\alpha_s^\top \psi(X))$, where $h(p) = \ln(p/(1 - p))$ is the logit link function (Mccullagh & Nelder, 1989).

Distributional changes between environments are accounted for by (1) through allowing the parameters $\alpha_s \in \mathbb{R}^z$ and $\beta_s \in \mathbb{R}^{z'}$ to change for different environments $s \in \{1, \dots, K\}$, in addition to changes in the covariate distribution $P(X \mid S = s)$. Changes in the parameters $(\alpha_s, \beta_s)$ correspond to shifts in the treatment assignment mechanism $\mathbb{E}[A \mid X, S = s] = \alpha_s^\top \psi(X)$ and outcome mechanism $\mathbb{E}[Y^a \mid X, S = s] = \beta_s^\top \phi(X, A = a)$; the feature representations $\psi(X)$ and $\phi(X, A)$ are considered to be fixed across environments. In practice, changes in the treatment assignment and outcome mechanisms are often expected. For example, if an unmeasured effect modifier exists, the outcome mechanism $\mathbb{E}[Y^a \mid X, S = s]$ will vary if the distribution of unmeasured effect modifiers differs across environments (Dahabreh et al., 2020a). Similarly, variation in the treatment assignment $\mathbb{E}[A \mid X, S = s]$ can be expected due to factors like differences in treatment policies across environments.

We will now pose the main assumption on how changes in the mechanism parameters $(\alpha_s, \beta_s)$ occur. Specifically, we assume there exists an unknown *prior distribution* $P(\alpha, \beta)$ that fulfills the following condition.

**Assumption 4.1.** The parameters $(\alpha_s, \beta_s) \sim P(\alpha, \beta)$ are drawn independently for each $s = 1, \dots, K$. Furthermore, the parameters are independent from each other such that $P(\alpha, \beta) = P(\alpha)P(\beta)$.

Following the principle of independent causal mechanisms, Assumption 4.1 states that the parameters $(\alpha_s, \beta_s)$ are *uninformative* of each other as they are sampled independently, and furthermore, that changing $\alpha$ has *no influence* on $\beta$, and vice versa. In the language of structural causal models, sampling $(\alpha_s, \beta_s)$ should be seen as independent soft

interventions on the distribution $P(X, A, Y \mid S = s)$. In the broader statistical context, Assumption 4.1 can also be related to hierarchical regression models with a prior independence assumption, see e.g. Gelman (2007, Chapter 11). Here, the independent sampling of the parameters $(\alpha_s, \beta_s)$ resembles the way hierarchical models account for variability between environments.

Finally, we will contrast our approach with falsification strategies based on transportability, which for instance would assume that the outcome mechanism $\mathbb{E}[Y^a \mid X, S = s]$ remains invariant across environments $S$. This assumption can be violated when unmeasured effect modifiers differ in distribution across environments, causing $\mathbb{E}[Y^a \mid X, S = s]$ to vary. In contrast, Assumption 4.1 does not require such invariance and explicitly allows this causal mechanism to vary. As a result, even when transportability fails to hold, Assumption 4.1 may still hold. We will later show that this makes our proposed falsification robust to violations of transportability, whereas transportability-based strategies may yield false positives: that is, incorrectly rejecting unconfoundedness despite the absence of unmeasured confounding. For a more detailed discussion of transportability-based falsification strategies, see Appendix A.

### 4.2. A testable implication under the independence assumption

To focus on the core ideas and limits of our falsification strategy, we assume the feature representations $\phi$ and $\psi$ are known up to some permutation and element-wise scaling. Moreover, to allow the use of standard estimation techniques, we will require the dimensionality of the feature representations to not be larger than any of the individual sample sizes among the different environments. We formalize these two conditions as follows.

**Assumption 4.2.** We have access to $\widetilde{\phi}(X) = C\phi(X)$ and $\widetilde{\psi}(X, A) = D\psi(X, A)$ where $C \in \mathbb{R}^{z \times z}$ and $D \in \mathbb{R}^{z' \times z'}$ are invertible matrices. The dimensionality of the feature representations $z, z' \in \mathbb{N}$ is finite and lower than the smallest sample size across environments, i.e, $z, z' < \min_s n_s$.

Our proposed falsification strategy will rely on estimating $\mathbb{E}[A \mid X, S = s]$ and $\mathbb{E}[Y \mid X, A, S = s]$ which under Assumption 3.1 corresponds to the true treatment and outcome mechanism. To estimate these conditional expectations, we employ two statistical working models $e_s(X) = \omega_s^\top \widetilde{\phi}(X)$ and $h_s(X, A) = \gamma_s^\top \widetilde{\psi}(X, A)$, respectively. Since we replaced the unknown feature representations $\{\phi, \psi\}$ with the observed feature representations $\{\widetilde{\phi}, \widetilde{\psi}\}$, the mechanism parameters $(\alpha_s, \beta_s)$ are replaced by $(\omega_s, \gamma_s)$. Our falsification strategy will test a statement equivalent to Assumption 4.1 but, again, substituting $(\alpha_s, \beta_s)$ with $(\omega_s, \gamma_s)$ as follows,

$$H_0 : P(\omega, \gamma) = P(\omega)P(\gamma) . \tag{2}$$

To understand how our falsification strategy will be centered around testing this null hypothesis, we begin by establishing the following key result (see Appendix B.1 for the proof).

**Theorem 4.3.** *Under the functional class described in* (1), *assumptions 3.1, 4.1 and 4.2, and with $e_s(X)$ and $h_s(X, A)$ being correctly specified models for $\mathbb{E}[A \mid X, S = s]$ and $\mathbb{E}[Y \mid X, A, S = s]$, we have that $H_0$ in* (2) *is true.*

The above theorem suggests that if we reject the null hypothesis $H_0$, it is likely because at least one of the conditions in the theorem is violated. While rejecting $H_0$ does not tell which condition in the theorem could be false, it still provides valuable information about the validity of the conditions in Assumption 3.1 which are necessary for treatment effect estimation. Before introducing our algorithm to statistically test $H_0$, we first explore a setting where violating Assumption 3.1 provably leads to the falsity of $H_0$.

### 4.3. Unmeasured confounding leads to mechanism dependencies

We examine a setting involving a linear causal model that includes both a main effect of treatment and interaction effects between treatment and covariates. While linearity may not always hold in real-world scenarios, this setting offers valuable insights into the conditions necessary to falsify causal assumptions in a multi-environment context.

To understand what effect an unmeasured confounder has on the independence of mechanisms, we introduce another unmeasured covariate $U \in \mathbb{R}$ as follows,

$$A = \alpha_s^\top \psi(X) + \alpha_s^{(U)} U + \varepsilon_A$$
$$Y^a = \beta_s^\top \phi(X, A = a) + \left( \beta_s^{(U)} + a\beta_s^{(AU)} \right) U + \varepsilon_Y \quad (3)$$

We let $\psi(X) = [1, X]^\top$ and $\phi(X, A) = [1, X, A, AX]^\top$ such that we can define the parameters $\alpha_s = [\alpha_s^{(0)}, \alpha_s^{(X)}]$ and $\beta_s = [\beta_s^{(0)}, \beta_s^{(X)}, \beta_s^{(A)}, \beta_s^{(AX)}]$. Throughout this example, we assume that $X \perp\!\!\!\perp U \mid S$.

The above causal model is partially observed because $U$ is an unmeasured covariate. If $U$ is a common cause to both the treatment $A$ and potential outcome $Y^a$, that is $\{\alpha_s^{(U)} \neq 0, \beta_s^{(U)} \neq 0\}$ and/or $\{\alpha_s^{(U)} \neq 0, \beta_s^{(AU)} \neq 0\}$, then we say that $U$ is an unmeasured confounder.

Whereas it is in general impossible to determine the presence of $U$, if we have correctly specified working models for $\mathbb{E}[A \mid X, S = s]$ and $\mathbb{E}[Y \mid A, X, S = s]$, we note that there exists dependencies between the observable parameters $\omega_s$ and $\gamma_s$ when $U$ is an unmeasured confounder (see Appendix B.2 for the proof).

**Lemma 4.4.** *Assume $U$ has a normal distribution with mean $\mu_s^U \in \mathbb{R}$ and standard deviation $\sigma_s^{(U)} \in \mathbb{R}^+$, and the noise variables $(\varepsilon_A, \varepsilon_Y)$ are normally distributed with mean zero*

*and standard deviations $\sigma^{(A)} \in \mathbb{R}^+$ and $\sigma^{(Y)} \in \mathbb{R}^+$. Consider the well-specified working models $e_s(X) = \omega_s^\top \widetilde{\phi}(X)$ and $h_s(X, A) = \gamma_s^\top \widetilde{\psi}(X, A)$ with $\widetilde{\phi}(X) = [1, X]^\top$ and $\widetilde{\psi}(X, A) = [1, X, A, AX, A^2]^\top$. Under the model in* (3) *with $U$ being an unmeasured confounder, we then have that the observable parameters are $\omega_s = \alpha_s + [\alpha_s^{(U)} \mu_s^{(U)}, 0]^\top$ and $\gamma_s = [\beta_s, 0]^\top + \Gamma_s$ where*

$$\Gamma_s = \delta_s \begin{bmatrix} -\beta_s^{(U)} \left( \frac{\alpha_s^{(0)} (\sigma_s^{(U)})^2}{\alpha_s^{(U)}} - \mu_s^{(U)} \left( \frac{\sigma_s^{(A)}}{\alpha_s^{(U)}} \right)^2 \right) \\ -\beta_s^{(U)} \frac{\alpha_s^{(X)} (\sigma_s^{(U)})^2}{\alpha_s^{(U)}} \\ \beta_s^{(U)} \frac{(\sigma_s^{(U)})^2}{\alpha_s^{(U)}} - \beta_s^{(AU)} \left( \frac{\alpha_s^{(0)} (\sigma_s^{(U)})^2}{\alpha_s^{(U)}} - \mu_s^{(U)} \left( \frac{\sigma_s^{(A)}}{\alpha_s^{(U)}} \right)^2 \right) \\ -\beta_s^{(AU)} \frac{\alpha_s^{(X)} (\sigma_s^{(U)})^2}{\alpha^{(U)}} \\ \beta_s^{(AU)} \frac{(\sigma_s^{(U)})^2}{\alpha_s^{(U)}} \end{bmatrix}$$

*and $\delta_s = \left( (\sigma_s^{(U)})^2 + \left( \frac{\sigma_s^{(A)}}{\alpha_s^{(U)}} \right)^2 \right)^{-1}$.*

The lemma, which holds for any $P(X \mid S = s)$, states that if $U$ is an unmeasured confounder then the observable parameters $(\gamma_s, \omega_s)$ have shared dependencies on the true parameters of the underlying data-generating process: the parameters $(\alpha_s^{(0)}, \alpha_s^{(X)}, \alpha_s^{(U)}, \mu_s^{(U)})$ appear in both the expressions of $\omega_s$ and $\gamma_s$.

The above results hold for both single-environment data ($K = 1$) and multi-environment data ($K > 1$), and does not rely on Assumption 4.1. Next, we show that our proposed falsification strategy allows us to detect the presence of the unmeasured confounder $U$ under certain conditions on the multi-environment structure when invoking Assumption 4.1 (see Appendix B.3 for the proof).

**Theorem 4.5.** *Under the assumptions stated in Lemma 4.4 and Assumption 4.1, if at least one of the following parameters $(\alpha_s^{(0)} \alpha_s^{(X)}, \alpha_s^{(U)}, \mu_s^{(U)})$ are i.i.d. sampled from a non-degenerate distribution for $s = 1, \ldots, K$, then $H_0$ is false if and only if $U$ is a confounder for all $s \in \{1, \ldots, K\}$.*

The theorem establishes that $H_0$ can be false due to violations of Assumption 3.1, which can be understood in terms of following statement: unmeasured confounding can create dependencies between observable parameters. The reason at least one of the parameters $(\alpha_s^{(0)} \alpha_s^{(X)}, \alpha_s^{(U)}, \mu_s^{(U)})$ must be sampled from a non-degenerate distribution is that this creates a statistical dependence between $\omega_s$ and $\gamma_s$ in the presence of an unmeasured confounder. Detecting this dependence becomes crucial for falsifying unconfoundedness.

The parameters $(\alpha_s^{(0)}, \alpha_s^{(X)} \alpha_s^{(U)}, \mu_s^{(U)})$ are related to the distributions $P(A \mid X, S = s)$ and $P(U \mid S = s)$. Thus, the non-degeneracy condition implies that falsifying Assumption 3.1 requires changes in either the treatment assignment or the distribution of the unmeasured confounder

across environments. This observation motivates the requirement of having multi-environment data: that is, without multiple environments there are no distributional changes that enable falsification to happen.

The same non-degeneracy condition was observed by Karlsson & Krijthe (2023) despite using a different formalism based on causal graphs. Their approach relies on identifying a specific d-separation via a conditional independence test between the treatment and outcome variables, which also allows for falsification of unconfoundedness across multiple environments. Their test can be interpreted as an indirect test of independence of causal mechanisms, as it operates solely on observed variable relationships. In contrast, our approach directly tests independence at the level of mechanism parameters. As a consequence, a further key difference is that their approach needs to make additional structural assumptions on the covariate distribution $P(X \mid S)$, while our theoretical results impose no such constraint.

## 5. Algorithm

We now introduce the Mechanism INdependent Test (MINT) algorithm, which operationalizes our falsification strategy for testing mechanism independence using data from multiple environments. We will use the following notation: for all environments $s = 1, \ldots, K$, we denote the observed data matrices as $\mathbf{A}_s = [A_1, \ldots, A_{n_s}]^\top$, $\mathbf{Y}_s = [Y_1, \ldots, Y_{n_s}]^\top$, $\widetilde{\Psi}_s = [\widetilde{\psi}(X_1), \ldots, \widetilde{\psi}(X_{n_s})]^\top$, and $\widetilde{\Phi}_s = [\widetilde{\phi}(X_1, A_1), \ldots, \widetilde{\phi}(X_{n_s}, A_{n_s})]^\top$.

The MINT algorithm can be divided into two steps: In the first stage, for all $s = 1, \ldots, K$, we estimate the parameters $(\omega_s, \gamma_s)$. The estimates are obtained through solving the least-squares problems $\widehat{\omega}_s = \arg\min_\omega \|\mathbf{A}_s - \widetilde{\Psi}_s \omega\|_2^2$ and $\widehat{\gamma}_s = \arg\min_\gamma \|\mathbf{Y}_s - \widetilde{\Phi}_s \gamma\|_2^2$ where $\|\cdot\|_2^2$ denotes the $l^2$-norm. We denote all estimated parameters as $\widehat{\omega} = [\widehat{\omega}_1, \ldots, \widehat{\omega}_K]$ and $\widehat{\gamma} = [\widehat{\gamma}_1, \ldots, \widehat{\gamma}_K]$. In the second stage, we perform a statistical independence test for the null hypothesis $H_0 : P(\omega, \gamma) = P(\omega)P(\gamma)$ using the estimated parameters $\widehat{\omega}$ and $\widehat{\gamma}$. If we accept $H_0$ then we should consider Assumption 3.1 and 4.1 to hold. On the other hand, if we reject $H_0$ then both assumptions are falsified jointly.

For the statistical independence test in the second stage, we study the co-variability of $(\gamma_s, \omega_s)$ across all environments by analyzing the covariance matrix $\Sigma = \mathrm{Cov}(\omega, \gamma)$. We propose using the test statistic $T = \sqrt{\sum_{i,j} \|\Sigma_{ij}\|^2}$ which is the Frobenius norm of the covariance matrix; crucially, this test statistic is always non-negative and $T = 0$ under $H_0$. The estimated test statistic becomes

$$\widehat{T}(\widehat{\omega}, \widehat{\gamma}) = \frac{1}{K} \sqrt{\sum_{i=1}^{z} \sum_{j=1}^{z'} \left[ \sum_{s=1}^{K} (\widehat{\omega}_{s,i} - \bar{\omega}_i)(\widehat{\gamma}_{s,j} - \bar{\gamma}_j) \right]^2}$$

where $\bar{\omega}_i = K^{-1} \sum_{s=1}^{K} \widehat{\omega}_{s,i}$ and $\bar{\gamma}_j = K^{-1} \sum_{s=1}^{K} \widehat{\gamma}_{s,j}$.

Lastly, we need to calibrate a rejection threshold $R$ such that we reject $H_0$ if $\widehat{T}(\widehat{\omega}, \widehat{\gamma}) > R$ while ensuring guarantees on the Type I error $\Pr(\widehat{T}(\widehat{\omega}, \widehat{\gamma}) > R \mid H_0) \leq \alpha$ for some $\alpha \in (0, 1)$. While this can be done using a permutation-based procedure with $M$ resamples, we have to take into account the uncertainty of the estimates from the model fitting in the first step of our algorithm.

To address this problem, we introduce an additional modification in the permutation-based calibration. Specifically, in the first step, we use bootstrapping and resample $M$ datasets with replacement to obtain estimates $\{(\widehat{\omega}^{(m)}, \widehat{\gamma}^{(m)})\}_{m=1}^{M}$. Then, for each $m = 1, \ldots, M$, we compute $T_m = T(\widetilde{\omega}^{(m)}, \widehat{\gamma}^{(m)})$ where $\widetilde{\omega}^{(m)}$ is a random permutation of $\widehat{\omega}^{(m)}$. Finally, we determine the rejection threshold as

$$R = \arg\max_{t \in (0,\infty)} \{t : M^{-1} \sum_{m=1}^{M} 1(T_m > t) \leq \alpha\},$$

where $1(T_m > t)$ equals 1 if $T_m > t$ and otherwise 0. Throughout the remainder of the paper, we let $M = 1000$. To highlight the importance of bootstrapping in the calibration, we present an ablation study in Appendix D.5 where we show that bootstrapping is essential for ensuring Type 1 errors remain below $\alpha$.

## 6. Experiments

We conducted a series of experiments to compare the proposed MINT algorithm with alternative baseline approaches. First, we investigate efficiency with respect to number of samples and number of environments. Next, we validate our theoretical findings by investigating necessary mechanism changes that allow for falsification. We then assessed the sensitivity of our algorithm to (mis)specification in its working models. Lastly, we evaluated all methods under more realistic conditions using semi-synthetic data based on the real-world Twins dataset (Almond et al., 2005), which includes birth data across different geographical locations used as environment labels.

We measured performance using the falsification rate (probability of falsification) and set the significance level $\alpha = 0.05$ to control Type 1 errors. In the absence of unmeasured confounding, the falsification rate reflects the Type I error rate and should remain below the significance level $\alpha = 0.05$. Conversely, in the presence of unmeasured confounding, the falsification rate corresponds to the statistical power of the algorithms, and thus, a higher rate is desirable. The code for reproducing our experiments is available at our GitHub repository.[1]

[1] https://github.com/RickardKarl/falsification-unconfoundedness

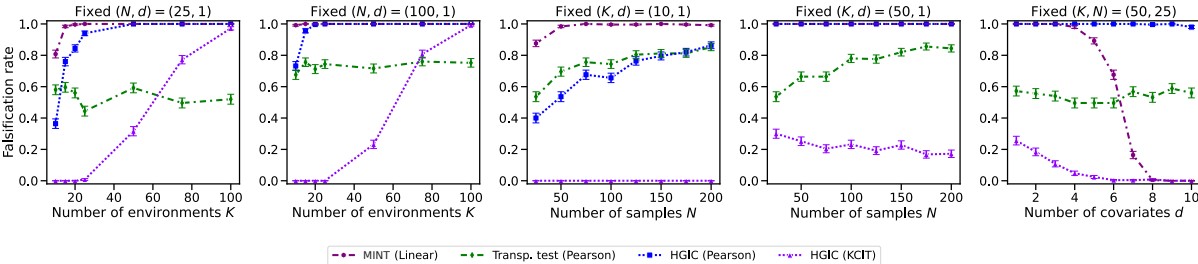

Figure 1: Comparison of falsification rate when varying either the number of environment $K$, the number of samples per environment $N$, or the number of observed covariates $d$. The error bars show the standard error over 250 repetitions.

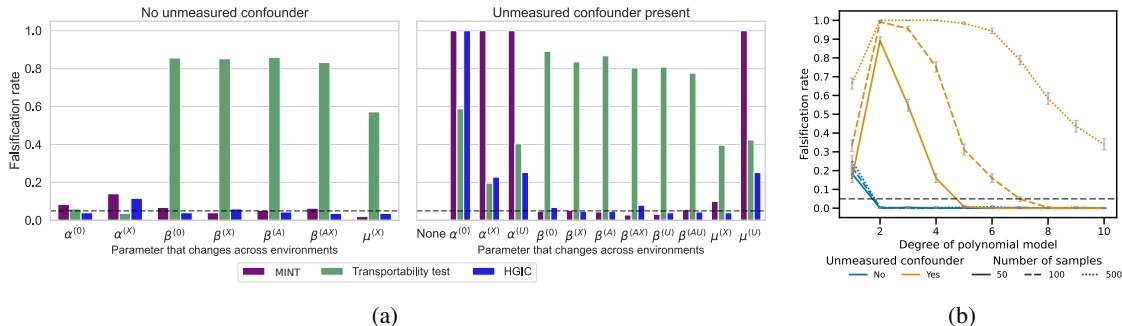

Figure 2: **(a):** Comparison of falsification rate when different mechanisms vary across the environment. The parameters on the x-axis correspond to those of the data-generating process in (3). **(b):** Our algorithm's performance is evaluated using polynomial basis functions as feature representation. The falsification rate is plotted against polynomial degree, with the true data-generating process including polynomials up to degree 2. The black dotted line in both figures correspond to the chosen significance level $\alpha = 0.05$. Average falsification rate and standard standard errors are reported over 250 repetitions. In the absence of unmeasured confounding, the falsification rate reflects the Type I error rate and should remain below the significance level $\alpha = 0.05$. Conversely, in the presence of unmeasured confounding, the falsification rate corresponds to the power of the test, and thus, a higher rate is desirable.

## 6.1. Baselines

We compare the proposed MINT algorithm to two baselines. The first, referred to as the transportability test, evaluates whether the independence $Y \perp\!\!\!\perp S \mid A, X$ holds, allowing for the joint falsification of Assumption 3.1 and the transportability condition $Y^a \perp\!\!\!\perp S \mid X$ (Dahabreh et al., 2020b). A detailed overview of transportability-based falsification strategies is provided in AppendixA. The second baseline is the hierarchical graph independence constraint (HGIC) approach (Karlsson & Krijthe, 2023). This approach tests a conditional independence statement based on a hierarchical description of the data. Unlike the transportability test, HGIC remains valid even when transportability conditions are violated, as it relies on an independence of causal mechanisms assumption similar to ours. This makes HGIC a strong candidate for comparison.

Since both baselines require selecting a conditional independence testing method, we evaluated them using either the Pearson partial correlation test, which is suitable for linear data, or the non-parametric kernel conditional independence

test (KCIT) (Zhang et al., 2011) with a radial basis function kernel, which is better suited for nonlinear data. For HGIC, we encountered some issues with KCIT that required minor modifications to the original implementation used by Karlsson & Krijthe (2023). These issues and the differences between our implementation and the original are discussed in detail in Appendix D.4.

## 6.2. Synthetic data

### 6.2.1. WHICH METHOD IS MOST EFFICIENT?

In the first experiment, we aimed to evaluate the efficiency of each method in a well-specified linear setting (see Appendix D.1 for more details on data generation). To make our method well-specified to the underlying data generating process, we used linear feature representations for MINT, and for the two baselines methods we used the partial Pearson correlation test which is suitable for conditional independence testing with linear data. Additionally, following Karlsson & Krijthe (2023), we tested HGIC using KCIT,

as it is also well-specified in this context.

We evaluated the falsification rate of each method under an unmeasured confounder while varying the number of environments $K$, the number of samples per environment $N$, or the number of observed covariates $d$, keeping the other factors fixed. The results are shown in Figure 1. When varying the number of environments $K$, our proposal MINT consistently outperformed HGIC. The transportability test was most effective when $K$ was small, though both MINT and HGIC showed higher falsification rates as $K$ increased. HGIC performed better with the Pearson test than with KCIT, highlighting the advantage of a parametric test in a well-specified setting. Increasing the number of samples $N$ improved falsification rates for all methods except HGIC with KCIT, although the gains were slower compared to increasing $K$. Finally, when varying $d$, HGIC with KCIT lost power the fastest, followed by MINT, while Pearson-based methods remained robust up to $d = 10$.

Additionally, in Appendix D.5, we confirm that all methods controlled Type 1 error in the absence of an unmeasured confounder, with the falsification rate remaining below the significance level $\alpha = 0.05$.

### 6.2.2. WHAT ARE NECESSARY MECHANISM CHANGES TO DETECT CONFOUNDING?

In the second experiment, we validated the theory behind our proposed MINT algorithm by generating various types of independent mechanism changes across the environments, following the model in (3) (see Appendix D.2 for details on data generation). We also applied the baseline methods to the same data to provide further insights into the necessary conditions for them to serve as a valid falsification strategy.

The different parameters on the x-axis in Figure 2a represent which of the parameters in (3) are varied across different environments, while all other parameters are kept fixed. This is done under both the absence and presence of unmeasured confounding. We observed that environmental changes in the parameters $(\alpha_s^{(0)}, \alpha_s^{(X)}, \alpha_s^{(U)}, \mu_s^{(U)})$, which influence either the treatment mechanism $\mathbb{E}[T \mid X, S = s]$ or the distribution of the unmeasured confounder $P(U \mid S = s)$, were sufficient for MINT to falsify unconfoundedness. This observation supports the claim we proved in Theorem 4.5.

Furthermore, both HGIC and the transportability test falsified under the same conditions when unmeasured confounders were present. However, the transportability test showed a notable issue with false positives in the absence of unmeasured confounding. The most likely explanation is that these false positives result from mechanism changes that violate the transportability condition, a key assumption for applying the transportability test.

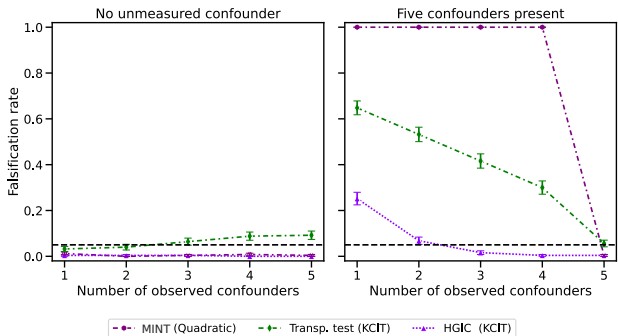

Figure 3: Comparison on the Twins semi-synthetic dataset. Average falsification rate and standard errors are reported over 250 repetitions; the black dotted line correspond to the chosen significance level $\alpha = 0.05$.

### 6.2.3. MODEL SPECIFICATION & PERFORMANCE

In the third experiment, we sampled data from a process with a polynomial basis function (see Appendix D.1 for more details on the data-generating process). We examined how changing the specification of the working models $e_s(X)$ and $h_s(X)$ in MINT affected its performance. The true polynomials in the data-generating process had a degree of 2, while MINT used a representation with polynomial basis functions with degrees ranging from 1 to 10. If the degree was set to 1, this introduced misspecification. For degrees of 2 or higher, the model was well-specified but became increasingly flexible as the polynomial degree increased.

As shown in Figure 2b, misspecified models led to an elevated false positive rate in the absence of unmeasured confounding. However, once the models were well-specified, false positives dropped below the nominal level $\alpha = 0.05$ even as model flexibility increased. When an unmeasured confounder was present, MINT successfully detected it, though its power (i.e., true positive rate) declined with higher model flexibility. We observed, however, that this reduction in power could be mitigated by increasing the number of samples per environment.

We also compared the transportability test and HGIC under both well-specified and misspecified settings. Using the Pearson partial correlation test on nonlinear data allowed us to assess their performance under misspecification. Similar to MINT, both exhibited higher false positive rates in the absence of unmeasured confounding when misspecified. Full results are provided in Table 2 in Appendix D.5.

### 6.3. Twins data

In the final experiment, we used data from twin births in the USA between 1989-1991 (Almond et al., 2005) to construct a multi-environment observational dataset with a known

causal structure. The environment corresponds to the birth state of each pair of twins. We generated treatment and outcome variables using the covariates from this dataset, providing a ground-truth causal structure to validate our methods while emulating realistic distributions with real-world covariates (see Appendix D.3 for details on dataset construction). The outcomes and treatment were modeled using a quadratic polynomial, and all methods were well-specified through either a quadratic polynomial feature representation or KCIT for conditional independence testing.

We examined a scenario with five confounders, varying the number of observed covariates from one to five. When all five confounders were observed, no unmeasured confounders remained; otherwise, some confounders were unmeasured. As a control, we repeated the experiment while varying the number of observed confounders but ensuring no unmeasured confounders. The results, shown in Figure 3, indicate that MINT outperforms both the transportability test and HGIC in terms of power. Additionally, when all five confounders were observed, MINT achieved the nominal falsification error below $\alpha = 0.05$.

## 7. Discussion

Our falsification strategy is not a silver bullet to detect unmeasured confounding. As we have demonstrated, our proposed algorithm is a joint falsification test that assesses both the conditions necessary for causal identification and the assumption of independent causal mechanisms. Thus, the limit to how informative this falsification test can be will depend on the plausibility of the independent causal mechanism assumption.

One reason our proposed algorithm performs well, especially compared to HGIC with KCIT, could be because of the parametric nature of our approach. While parametric assumptions can be incorporated into both HGIC and the transportability test by selecting an appropriate conditional independence test, such as the Pearson test used in our experiment, our algorithm encodes these assumptions differently. Specifically, it explicitly incorporates the parametric assumptions for both the treatment and outcome models. Interestingly, this approach aligns more closely with the common practice of specifying both models when estimating treatment effects in observational studies.

A drawback of relying on parametric assumptions for the treatment and outcome models is the increased Type 1 error under misspecification. This happened to our algorithm when $\{\tilde{\psi}, \tilde{\phi}\}$ were misspecified. So far, we have assumed these representations are known a priori. To mitigate the risk of misspecification, one strategy is to construct feature representations that apply a broad set of transformations to the observed covariates. This ensures the representation is suf-

ficiently expressive to capture the underlying relationships in the data. However, increasing the richness of the feature representation introduces a trade-off: while it reduces the risk of misspecification, it can also decrease statistical power due to increased model complexity. This trade-off was evident in our experiments (Figure 2), where increasing model complexity lead to a reduction in power of the test.

Hence, a key future direction is to address the case where $\{\tilde{\psi}, \tilde{\phi}\}$ are unknown and attempt to learn them from data. In this case, we have shown that it would be sufficient to learn them up to some permutation and element-wise scaling. Alternatively, we could attempt to use more flexible (implicit) feature representations through the use of kernel methods (Schölkopf & Smola, 2002). Whereas further work is needed to adapt our theory to a kernelized algorithm, we provide a sketch as a starting point for such an approach in Appendix C.

## 8. Conclusion

We propose novel algorithmic ideas to directly exploit observed dependencies in causal mechanisms for falsification of the assumptions necessary for causal effect identification. Specifically, we propose a two-stage algorithm that can be applied to multi-environment data. Although there are no universal solutions for addressing untestable assumptions in causal inference, we believe that our proposal has an important place in evaluating the necessary conditions to enable more trustworthy causal conclusions.

## Impact Statement

Causal inference plays a crucial role in real-world decision-making, underpinning fields from medicine to public policy. While our work aims to enhance the reliability and safety of causal inference methods, it remains an early-stage development. We stress the importance of careful implementation in collaboration with domain experts, particularly in high-stakes settings.

## Acknowledgements

Research reported in this work was was facilitated by the computational resources and support of the Delft AI Cluster (DAIC) at TU Delft. We also thank our anonymous reviewers for their helpful comments and input.

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

# A. Falsification with transportability conditions

A common way of using data from multiple environments to falsify the validity of Assumption 3.1 is to assume a transportability condition that relates the different environments to each other. One of the most common way of formalizing the transportability condition is as follows.

**Assumption A.1** (Conditional exchangeability between environments)**.** We assume for all $a \in \mathcal{A}$, $Y^a \perp\!\!\!\perp S \mid X$.

Other variations of the transportability condition is to assume conditional mean exchangeability $\mathbb{E}[Y^a \mid X, S = s] = \mathbb{E}[Y^a \mid X]$ or that an effect measure such as the conditional average treatment effect $\mathbb{E}[Y^1 - Y^0 \mid X, S = s] = \mathbb{E}[Y^1 - Y^0 \mid X]$ is transportable (Colnet et al., 2024).

It is well-known that Assumption 3.1 and A.1 together have a testable implication in the law of the observed data, see e.g. Dahabreh et al. (2020b). More specifically, Assumption 3.1 and A.1 together imply that the following conditional independence must be true

$$Y \perp\!\!\!\perp S \mid X, A \ . \tag{4}$$

Testing (4) can be done with any suitable conditional independence test and efficient procedures also exists for testing implications from the other variations of the transportability condition, see e.g. Hussain et al. (2023). However, the underlying premise is always the same: if one would conclude that (4) does not hold, then this could be due to either a violation of Assumption 3.1 or A.1. Thus, if one believes that Assumption A.1 must hold yet observes (4) to be false, that means that Assumption 3.1 must be violated in at least one of the environments. This argument becomes particularly strong if treatment has been randomized in one of the environments since any difference between the environments is more likely to be explained by an unmeasured confounder in the remaining environments with observational data. However, Assumption A.1 itself can be controversial as it is also untestable. More specifically, it would be violated if there are unmeasured so-called effect modifiers which are covariates that differ in distribution between environments and modulate treatment heterogeneity. Effect modifiers are distinct from confounders as effect modifiers only need to be a cause of the outcome of interest. Thus, confounders can be effect modifiers but not vice versa, meaning that we often might expect to have unmeasured effect modifiers present even where are no unmeasured confounders.

The primary distinction between our falsification strategy and a transportability-based falsification strategy lies in the assumption our strategy relies on: instead of using Assumption A.1, our strategy employs Assumption 4.1 to derive an alternative jointly testable implication. This comparison also highlights their underlying similarity: both strategies aim to combine two untestable assumptions to generate a testable implication, enabling the joint falsification of these otherwise untestable assumptions.

# B. Proofs

## B.1. Proof of Theorem 4.3

*Proof.* Using the conditions from Assumption 3.1 and that $h_s(X) = \gamma_s^\top \widetilde{\phi}(X, A = a)$ is a correctly specified model for $\mathbb{E}[Y \mid A, X, S = s]$, we can write

$$
\begin{aligned}
\beta_s^\top \phi(X, A = a) &= \mathbb{E}[Y^a \mid X, S = s] \\
&= \mathbb{E}[Y^a \mid X, A = a, S = s] && Y^a \perp\!\!\!\perp A \mid (X, S = s) \\
&= \mathbb{E}[Y \mid X, A = a, S = s] && A = a \Rightarrow Y^a = Y \\
&= \gamma_s^\top \widetilde{\phi}(X, A = a) \ .
\end{aligned}
$$

Because we assumed $\widetilde{\phi}(X, A = a) = C\phi(X, A = a)$ for some invertible matrix $C$ (Assumption 4.2), it follows for $s = 1, \ldots, K$ that $\gamma_s = (C^{-1})^\top \beta_s$. Furthermore, using that $e_x(X) = \omega_s^\top \widetilde{\psi}(X, A = a)$ is a correctly specified model for $\mathbb{E}[A \mid, X, S = s]$ and $\widetilde{\psi}(X, A) = D\psi(X, A)$ for some invertible matrix $D$ (again, Assumption 4.2), it follows using similar arguments that $\omega_s = (D^{-1})^\top \alpha_s$ for $s = 1, \ldots, K$. To conclude the proof, using Assumption 4.1 which states that there exists a distribution $P(\alpha, \beta) = P(\alpha)P(\beta)$, we observe that $(\omega_s, \gamma_s)$ are distributed according to a distribution defined as $P(\omega, \gamma) := P((C^{-1})^\top \alpha, (D^{-1})^\top \beta)$. It is well-known that if $\alpha_s$ and $\beta_s$ are independent random variables, then their transformations $(C^{-1})^\top \alpha_s$ and $(D^{-1})^\top \beta_s$ are also independent; see Grimmett & Stirzaker (2020, Chapter 4.2). Thus, we have $P(\alpha, \beta) = P(\alpha)P(\beta) \iff P(\omega, \gamma) = P(\omega)P(\gamma)$. $\square$

## B.2. Proof of Lemma 4.4

Before we can prove the lemma, we need the following auxiliary result.

**Lemma B.1.** *Consider two Normal probability densities* $f_1(x) = \frac{1}{\sqrt{2\pi\sigma_1^2}} \exp\left(-\frac{1}{2\sigma_1^2}(x-\mu_1)^2\right)$ *and* $f_2(x) = \frac{1}{\sqrt{2\pi\sigma_2^2}} \exp\left(-\frac{1}{2\sigma_2^2}(x-\mu_2)^2\right)$ *with* $\sigma_1, \sigma_2 > 0$. *We then have that the product of the densities is* $f_1(x) \cdot f_2(x)$ *is proportional to a Normal density* $\frac{1}{\sqrt{2\pi\sigma_{12}^2}} \exp\left(-\frac{1}{2\sigma_{12}^2}(x-\mu_{12})^2\right)$ *where*

$$\mu_{12} = \frac{\mu_1\sigma_2^2 + \mu_2\sigma_1^2}{\sigma_1^2 + \sigma_2^2} \text{ and } \sigma_{12}^2 = \frac{\sigma_1^2\sigma_2^2}{\sigma_1^2 + \sigma_2^2}.$$

*Proof.* Note that

$$f_1(x) \cdot f_2(x) = \frac{1}{2\pi\sigma_1\sigma_2} \exp\left(-\frac{1}{2}\underbrace{\left[\frac{(x-\mu_1)^2}{\sigma_1^2} + \frac{(x-\mu_2)^2}{\sigma_2^2}\right]}_{Q}\right),$$

where the expression inside the exponential function can be written as

$$\begin{aligned}
Q &= \frac{\left(\sigma_1^2 + \sigma_2^2\right)x^2 - 2\left(\mu_1\sigma_2^2 + \mu_2\sigma_1^2\right)x + \mu_1^2\sigma_2^2 + \mu_2^2\sigma_1^2}{\sigma_1^2\sigma_2^2} \\
&= \frac{x^2 - 2\frac{\left(\mu_1\sigma_2^2+\mu_2\sigma_1^2\right)}{\left(\sigma_1^2+\sigma_2^2\right)}x + \frac{\mu_1^2\sigma_2^2+\mu_2^2\sigma_1^2}{\left(\sigma_1^2+\sigma_2^2\right)}}{\frac{\sigma_1^2\sigma_2^2}{\left(\sigma_1^2+\sigma_2^2\right)}} \\
&= \frac{\left(x - \frac{\left(\mu_1\sigma_2^2+\mu_2\sigma_1^2\right)}{\left(\sigma_1^2+\sigma_2^2\right)}\right)^2}{\frac{\sigma_1^2\sigma_2^2}{\left(\sigma_1^2+\sigma_2^2\right)}} + \frac{\frac{\left(\mu_1\sigma_2^2+\mu_2\sigma_1^2\right)}{\left(\sigma_1^2+\sigma_2^2\right)} + \frac{\mu_1^2\sigma_2^2+\mu_2^2\sigma_1^2}{\left(\sigma_1^2+\sigma_2^2\right)}}{\frac{\sigma_1^2\sigma_2^2}{\left(\sigma_1^2+\sigma_2^2\right)}}.
\end{aligned}$$

As the second term on the last line is independent of $x$, we finish the proof by observing that $f_1(x) \cdot f_2(x)$ is up to some constant proportional to

$$\frac{1}{\sqrt{2\pi\frac{\sigma_1^2\sigma_2^2}{\sigma_1^2+\sigma_2^2}}} \exp\left(-\frac{1}{2}\frac{\left(x - \frac{\left(\mu_1\sigma_2^2+\mu_2\sigma_1^2\right)}{\left(\sigma_1^2+\sigma_2^2\right)}\right)^2}{\frac{\sigma_1^2\sigma_2^2}{\left(\sigma_1^2+\sigma_2^2\right)}}\right).$$

$\square$

Next, we proceed with the proof of lemma 4.4.

*Proof.* To simplify notation, we will drop the subscript $s$ for all parameters. We start with $\mathbb{E}[A \mid X = x, S = s]$, which can be written as

$$\begin{aligned}
\mathbb{E}[A \mid X, S = s] &= \mathbb{E}[\alpha^{(0)} + \alpha^{(X)}X + \alpha^{(U)}U + \varepsilon_A \mid X, S = s] \\
&= \alpha^{(0)} + \alpha^{(X)}X + \alpha^{(U)}\mathbb{E}[U \mid X, S = s] + \underbrace{\mathbb{E}[\varepsilon_A \mid X, S = s]}_{=0} \\
&\overset{(a)}{=} \alpha^{(0)} + \alpha^{(X)}X + \alpha^{(U)}\mu^{(U)} \\
&= \left(\begin{bmatrix}\alpha_0 \\ \alpha_X\end{bmatrix} + \begin{bmatrix}\alpha^{(U)}\mu^{(U)} \\ 0\end{bmatrix}\right)^\top \begin{bmatrix}1 \\ X\end{bmatrix}
\end{aligned}$$

where in (a) we use that $X \perp\!\!\!\perp U \mid (S = s)$ meaning that $\mathbb{E}[U \mid X, S = s]\mathbb{E}[U \mid S = s] = \mu^{(U)}$.

Next, we continue with $\mathbb{E}[Y \mid X, A, S = s]$, which can be expressed as

$$
\begin{aligned}
\mathbb{E}\Big[&\beta^{(0)} + \beta^{(X)}X + \beta^{(U)}U + A\left(\beta^{(A)} + \beta^{(AX)}X + \beta^{(AU)}U\right) + \varepsilon_Y \mid X, A, S = s\Big] \\
&= \beta^{(0)} + \beta^{(X)}X + A\left(\beta^{(A)} + \beta^{(AX)}X\right) \\
&\quad + \left(\beta^{(U)} + A\beta^{(AU)}\right)\mathbb{E}[U \mid X, A, S = s] + \underbrace{\mathbb{E}[\varepsilon_Y \mid X, A, S = s]}_{=0}.
\end{aligned}
\tag{5}
$$

To evaluate the conditional expectation $\mathbb{E}[U \mid X, A, S = s]$, we use Bayes rule to rewrite the probability density function

$$
f(U \mid X, A, S) = \frac{f(A \mid X, U, S)f(U \mid X, S)}{f(A \mid X, S)}.
$$

Firstly, note that $f(U \mid X, S) = f(U \mid S)$ follows from that $X \perp\!\!\!\perp U \mid (S = s)$. The density $f(U \mid S)$ is corresponds to the density of $N(\mu^{(U)}, (\sigma^{(U)})^2)$. Secondly, we can express $f(A \mid X, U, S)$ differently by exploiting that $A = \alpha^{(0)} + \alpha^{(X)}X + \alpha^{(U)}U + \varepsilon_A$ as follows,

$$
\begin{aligned}
f(A = a \mid X = x, U = u, S = s) &= f(\varepsilon_A = a - \alpha^{(0)} - \alpha^{(X)}x - \alpha^{(U)}u \mid S = s) \\
&\stackrel{(b)}{=} \frac{1}{\sqrt{2\pi(\sigma^{(A)})^2}}\exp\left(-\frac{1}{2(\sigma^{(A)})^2}(a - \alpha^{(0)} - \alpha^{(X)}x - \alpha^{(U)}u)^2\right) \\
&\stackrel{(c)}{=} \frac{1}{\sqrt{2\pi(\sigma^{(A)})^2}}\exp\left(-\frac{1}{2\left(\frac{\sigma^{(A)}}{\alpha^{(U)}}\right)^2}\left((\alpha^{(U)})^{-1}\left(a - \alpha^{(0)} - \alpha^{(X)}x\right) - u\right)^2\right)
\end{aligned}
$$

where (b) follows from that $\varepsilon_A \mid (S = s) \sim N(0, (\sigma^{(A)})^2)$. In (c) we reshuffle terms to explicitly break out $u$ inside the exponential function. From inspecting the expression on the last line, we note that it looks like an unnormalized probability density function w.r.t $u$ for a Normal distribution. If we rescale $f(A = a \mid X = x, U = u, S = s)$ by $\frac{1}{\sigma^{(U)}}$, we obtain an probability density function w.r.t $u$ for the Normal distribution $N\left((\alpha^{(U)})^{-1}\left(a - \alpha^{(0)} - \alpha^{(X)}x\right), \left(\frac{\sigma^{(A)}}{\alpha^{(U)}}\right)^2\right)$. This observation together with the results from lemma B.1 allows us to show that the product of densities $f(A \mid X, U, S)f(U \mid X, S)$ also corresponds to an unnormalized, scaled probability density function of a Normal distribution with mean equal to

$$
\frac{(\alpha^{(U)})^{-1}\left(a - \alpha^{(0)} - \alpha^{(X)}x\right)(\sigma^{(U)})^2 + \mu^{(U)}\left(\frac{\sigma^{(A)}}{\alpha^{(U)}}\right)^2}{(\sigma^{(U)})^2 + \left(\frac{\sigma^{(A)}}{\alpha^{(U)}}\right)^2}.
\tag{6}
$$

Since we can re-normalize $f(A \mid X, U, S)f(U \mid X, S)$ with $1/f(A \mid X, S)$, we have that the conditional expectation

$$
\mathbb{E}[U \mid X, A, S = s] = \int u \frac{f(A \mid X, U = u, S = s)f(U = u \mid X, S = s)}{f(A \mid X, S = s)}du
$$

is equal to (6).

Plugging (6) back into (5), we have that

$$
\begin{aligned}
\mathbb{E}[Y \mid X, A, S = s] &= \beta^{(0)} + \beta^{(X)}X + A\left(\beta^{(A)} + \beta^{(AX)}X\right) \\
&\quad + \left(\beta^{(U)} + A\beta^{(AU)}\right)\left(\frac{(\alpha^{(U)})^{-1}\left(A - \alpha^{(0)} - \alpha^{(X)}X\right)(\sigma^{(U)})^2 + \mu^{(U)}\left(\frac{\sigma^{(A)}}{\alpha^{(U)}}\right)^2}{(\sigma^{(U)})^2 + \left(\frac{\sigma^{(A)}}{\alpha^{(U)}}\right)^2}\right).
\end{aligned}
$$

To conclude the proof, we simplify the above expression to the form $\mathbb{E}[Y \mid X, A, S = s] = \gamma^\top [1, X, A, AX, A^2]^\top$ where

$$
\gamma = \begin{bmatrix} \beta^{(0)} \\ \beta^{(X)} \\ \beta^{(A)} \\ \beta^{(AX)} \\ 0 \end{bmatrix} + \delta \begin{bmatrix} -\beta^{(U)} \left( \frac{\alpha^{(0)} (\sigma^{(U)})^2}{\alpha^{(U)}} - \mu^{(U)} \left( \frac{\sigma^{(A)}}{\alpha^{(U)}} \right)^2 \right) \\ -\beta^{(U)} \frac{\alpha^{(X)} (\sigma^{(U)})^2}{\alpha^{(U)}} \\ \beta^{(U)} \frac{(\sigma^{(U)})^2}{\alpha^{(U)}} - \beta^{(AU)} \left( \frac{\alpha^{(0)} (\sigma^{(U)})^2}{\alpha^{(U)}} - \mu^{(U)} \left( \frac{\sigma^{(A)}}{\alpha^{(U)}} \right)^2 \right) \\ -\beta^{(AU)} \frac{\alpha^{(X)} (\sigma^{(U)})^2}{\alpha^{(U)}} \\ \beta^{(AU)} \frac{(\sigma^{(U)})^2}{\alpha^{(U)}} \end{bmatrix} .
$$

and $\delta = \left( (\sigma^{(U)})^2 + \left( \frac{\sigma^{(A)}}{\alpha^{(U)}} \right)^2 \right)^{-1}$. $\qquad\qquad\square$

## B.3. Proof of Theorem 4.5

Before proving the theorem, we present the following result that we will need later.

**Lemma B.2.** *Under the data generating process in* (3) *and* $\{\alpha^{(U)} = 0\}$, *we have that* $\mathbb{E}[Y \mid X, A, S = s] = \alpha_s^\top [1, X]^\top$ *and* $\mathbb{E}[Y \mid X, A, S = s] = \gamma_s^\top [1, X, A, AX, A^2]^\top$ *where*

$$
\alpha_s = \begin{bmatrix} \alpha_s^{(0)} \\ \alpha_s^{(X)} \end{bmatrix}, \quad \gamma_s = \begin{bmatrix} \beta_s^{(0)} \\ \beta_s^{(X)} \\ \beta_s^{(A)} \\ \beta_s^{(AX)} \\ 0 \end{bmatrix} + \begin{bmatrix} \beta_s^{(U)} \mu_s^{(U)} \\ 0 \\ \beta_s^{(AU)} \mu_s^{(U)} \\ 0 \\ 0 \end{bmatrix} .
$$

*On the other hand, if instead of* $\{\alpha^{(U)} = 0\}$ *we have that* $\{\beta^{(U)} = 0, \beta^{(AU)} = 0\}$ *then*

$$
\alpha_s = \begin{bmatrix} \alpha_s^{(0)} + \alpha_s^{(U)} \mu_s^{(U)} \\ \alpha_s^{(X)} \end{bmatrix}, \quad \gamma_s = \begin{bmatrix} \beta_s^{(0)} \\ \beta_s^{(X)} \\ \beta_s^{(A)} \\ \beta_s^{(AX)} \\ 0 \end{bmatrix} .
$$

*Proof.* For the first case with $\{\alpha^{(U)} = 0\}$, we have that $\mathbb{E}[A \mid X, S = s] = \mathbb{E}[\alpha_s^{(0)} + \alpha_s^{(X)} X + \varepsilon_A \mid S = s] = \alpha_s^{(0)} + \alpha_s^{(X)} X$. Further, we can show

$$
\begin{aligned}
\mathbb{E}[Y \mid X, A, S = s] &= \beta_s^{(0)} + \beta_s^{(X)} X + \beta_s^{(A)} A + \beta_s^{(AX)} AX + \mathbb{E}[\beta_s^{(U)} U + \beta_s^{(AU)} AU \mid X, A, S = s] \\
&= \beta_s^{(0)} + \beta_s^{(X)} X + \beta_s^{(A)} A + \beta_s^{(AX)} AX + \beta_s^{(U)} \mu_s^{(U)} + \beta_s^{(AU)} \mu_s^{(U)} A
\end{aligned}
$$

where the second equality from that $U \perp\!\!\!\perp X \mid A, S = s$ holds in (3) if $\{\alpha^{(U)} = 0\}$. This concludes the first case.

For the second case with $\{\beta^{(U)} = 0, \beta^{(AU)} = 0\}$, it follows from the above equation that

$$
\mathbb{E}[Y \mid X, A, S = s] = \beta_s^{(0)} + \beta_s^{(X)} X + \beta_s^{(A)} A + \beta_s^{(AX)} AX .
$$

Meanwhile, for the treatment mechanism we now instead have that

$$
\begin{aligned}
\mathbb{E}[A \mid X, S = s] &= \alpha_s^{(0)} + \alpha_s^{(X)} X + \mathbb{E}[\alpha_s^{(U)} U \mid S = s] \\
&= \alpha_s^{(0)} + \alpha_s^{(X)} X + \alpha_s^{(U)} \mu_s^{(U)} .
\end{aligned}
$$

$\qquad\qquad\square$

Now, we can proceed with the proof of the Theorem 4.5.

*Proof.* Throughout the proof, we define $\phi(X) = [1, X]^\top$ and $\psi(X) = [1, X, A, AX, A^2]^\top$. We will show that regardless of the presence of the confounder $U$, we can write $\mathbb{E}[A \mid X, S = s] = \omega_s^\top \phi(X)$ and $\mathbb{E}[Y \mid X, A, S = s] = \gamma_s^\top \psi(X, A)$ for some $(\omega_s, \gamma_s)$ and only if and only if $U$ is a confounder will $\omega_s \not\perp\!\!\!\perp \gamma_s$ under some conditions on the parameters $(\alpha_s^{(0)}, \alpha_s^{(X)}, \alpha_s^{(U)}, \mu_s^{(U)})$.

Note that under Assumption 4.1, it follows that there exists a distribution $(\omega_s, \gamma_s) \sim P(\omega, \gamma)$ since the parameters $(\omega_s, \gamma_s)$ are directly dependent on the parameters $(\alpha_s, \beta_s) \sim P(\alpha, \beta)$, for $s = 1, \ldots, K$. To determine if $P(\omega, \gamma) = P(\omega)P(\gamma)$, we have to determine whether $\omega_s$ and $\gamma_s$ both depend on the same parameters from the underlying data-generating process and under what conditions this can create statistical dependencies between $\omega_s$ and $\gamma_s$.

**No unmeasured confounding present**   First, consider the condition that $U$ is not a confounder. There are three cases for which this happens: (1) we have $\{\alpha^{(U)} = 0\}$, (2) we have $\{\beta^{(U)} = 0, \beta^{(AU)} = 0\}$, and (3) we have both $\{\alpha^{(U)} = 0\}$ and $\{\beta^{(U)} = 0, \beta^{(AU)} = 0\}$. For case (1) and (2), it follows immediately from lemma B.2 that $\omega_s$ and $\gamma_s$ have no shared parameters. For the final case (3), it is easy to see that $\omega_s = [\alpha_s^{(0)}, \ \alpha_s^{(X)}]^\top$ and $\gamma_s^\top = [\beta_s^{(0)}, \ \beta_s^{(X)}, \ \beta_s^{(A)}, \ \beta^{(AX)}, \ 0]$ where, again, there are no shared parameters between $\omega_s$ and $\gamma_s$. Thus, we can conclude that under all of the cases when $U$ is not a confounder, $(\omega_s, \gamma_s)$ have no shared parameter and thus $\omega_s \perp\!\!\!\perp \gamma_s$.

**Unmeasured confounding present**   Next, consider the condition that $U$ is a confounder. It follows from lemma 4.4 that if $U$ is a confounder, then both $\omega_s$ and $\gamma_s$ depend on the parameters $(\alpha_s^{(0)}, \alpha_s^{(X)}, \alpha_s^{(U)}, \mu_s^{(U)})$. Thus, if any of the parameters $(\alpha_s^{(0)}, \alpha_s^{(X)} \alpha_s^{(U)}, \mu_s^{(U)})$ vary across different environments, which happens if we assume there exist a non-degenerate distribution for at least of one these parameters, it follows that $\omega_s \not\perp\!\!\!\perp \gamma_s$. This concludes the proof. $\square$

## C. Extension to implicit feature representations

In this section, we provide a sketch for replacing the features representation $\{\widetilde{\phi}, \widetilde{\psi}\}$ in our falsification algorithm with an implicit feature representation through the use of kernel methods (Schölkopf & Smola, 2002). More specifically, we let $\widetilde{\phi}(x)$ be the implicit feature representation whose inner product is given by the kernel $k(x, x) = \langle \widetilde{\phi}(x), \widetilde{\phi}(x) \rangle_{\mathcal{H}}$ defined on $\mathcal{X}$ with corresponding RKHS $\mathcal{H}$ and, similarly, let $\widetilde{\psi}(x, a)$ be the implicit feature representation with inner product given by $h((x, a), (x, a)) = \langle \widetilde{\psi}(x, a), \widetilde{\psi}(x, a) \rangle_{\mathcal{G}}$ defined on $\mathcal{X} \otimes \mathcal{A}$ with corresponding RKHS $\mathcal{G}$.

With some minor modifications, we can run our falsification algorithm without having to compute $\widetilde{\phi}(x)$ and $\widetilde{\psi}(x)$. To illustrate this, we impose the restriction that we use the same number of observations from each environment, denoted with $n$. We shall focus on estimators given by $\widehat{\omega}_s = \arg\min_\omega = \|\mathbf{A}_s - \Phi_s \omega\|_2^2 + \lambda_1 \|\omega\|_2$ and $\widehat{\gamma}_s = \arg\min_\omega = \|\mathbf{Y}_s - \Psi_s \gamma\|_2^2 + \lambda_2 \|\gamma\|_2$ for some constants $\lambda_1, \lambda_2 > 0$. The above optimization problems correspond to kernel ridge regression problem, for which it is well-known that the estimators have a closed form solution, namely

$$\widehat{\omega}_s = (K_s + n\lambda_1 I_n)^{-1} \mathbf{A}_s \text{ and } \widehat{\gamma}_s = (H_s + n\lambda_2 I_n)^{-1} \mathbf{Y}_s$$

where $K_s = k(\mathbf{X}_s, \mathbf{X}_s)$ and $H_s = h((\mathbf{X}_s, \mathbf{A}_s), (\mathbf{X}_{s'}, \mathbf{A}_{s'}))$ are the Gram matrices. The above estimators are however not always be computable since they can, depending on the choice of kernel, be infinite-dimensional. This also makes it infeasible to directly compute the covariance matrix $\Sigma = \text{Cov}(\omega, \gamma)$. However, it is luckily still possible to compute the Frobenius norm $\|\Sigma\|_2$. Using results from lemma 1 in Gretton et al. (2005), we can rewrite $\|\Sigma\|_2 = \mathbb{E}_{P(\omega, \gamma)}[\omega^\top \omega \, \gamma^\top \gamma] + \mathbb{E}_{P(\omega)}[\omega^\top \omega] \mathbb{E}_{P(\gamma)}[\gamma^\top \gamma] - 2\mathbb{E}_{P(\omega, \gamma)}[\mathbb{E}_{P(\omega)}[\omega^\top \omega] \mathbb{E}_{P(\gamma)}[\gamma^\top \gamma]]$. From this equality, it follows that we can compute $\|\Sigma\|_2$ by inspecting the inner products $\widehat{\omega}_s^\top \widehat{\omega}_{s'}$ and $\widehat{\gamma}_s^\top \widehat{\gamma}_{s'}$ for all $s, s' \in \{1, \ldots, K\}$. Interestingly, these inner products can be computed as follows

$$\widehat{\omega}_s^\top \widehat{\omega}_{s'} = \mathbf{A}_s^\top (K_s^\top + n\lambda_1 I_n)^{-1} (K_{s'} + n\lambda_1 I_n)^{-1} \mathbf{A}_{s'}$$
$$\widehat{\gamma}_s^\top \widehat{\gamma}_{s'} = \mathbf{Y}_s^\top (H_s^\top + n\lambda_1 I_n)^{-1} (H_{s'} + n\lambda_1 I_n)^{-1} \mathbf{Y}_{s'}$$

which means that $\|\Sigma\|_2$ can be computed and we can in principle statistically test for independence of $\omega$ and $\gamma$ with some implicit feature representations $\{\widetilde{\phi}, \widetilde{\psi}\}$. For future work, it remains unknown how to best implement this algorithm and investigate how our theory needs to be modified for it.

# D. Experimental details

### D.1. Sampling from data-generating process with polynomial basis functions

We generated observational datasets as follows. For each environment $s = 1, \ldots, K$, we obtain $i = 1, \ldots, N$ individuals by first sampling a set of $d$-dimensional covariates according to $X_i \sim N(\mu_s^{(X)}, \frac{1}{\sqrt{d}}\Sigma)$ with the mean $\mu_s^{(X)} \in \mathbb{R}^d \sim N(\mathbf{0}, \frac{1}{4}\mathbf{I}_d)$ where $\mathbf{I}_d$ was a $d \times d$ identity matrix and the covariance matrix $\Sigma$ of shape $d \times d$ had its diagonal elements set to 2 and its off-diagonal elements set to 0.1. Thereafter, we sampled the treatment $T_i$ and outcome $Y_i$ according to (1) with the features representations $\psi(X) = [1, X_1, \ldots, X_d, X_1^p, \ldots, X_d^p]^\top$ and $\phi(X, A) = [1, X_1, \ldots, X_d, X_1^p, \ldots, X_d^p, A]^\top$ being polynomial basis functions of degree $p$. The noise variables $\varepsilon_A$ and $\varepsilon_Y$ were mean-zero Normal distributed with their standard deviation set to 0.5. Each element in $\alpha_s$ where sampled uniformly from the set $\{-1, 1\}$ while each element in $\beta_s$ was set to 1, except for the elements in $\alpha_s$ corresponding to the intercepts which were sampled according to $N(0, 1)$. Only the intercept elements were resampled for each new environment, whereas the remaining coefficients in $(\alpha_s, \beta_s)$ where kept fixed for all environments. When introducing an unmeasured confounder, we additionally sampled an one-dimensional covariate $U_i \sim N(\mu_s^{(U)}, 2)$ with its mean $\mu_s^{(U)} \sim N(0, 1)$. Then, we added $U_i$ directly to $T_i$ and $Y_i$. For simplicity, we let each environment have the same number of samples $N = n_1 = \cdots = n_K$ even though all methods also work if the number of samples per environment differ.

### D.2. Sampling from data-generating process in (3)

We sampled a multi-environment dataset with $K = 250$ environments and 1000 samples per environment according to the data-generating process described in Section 4.3:

$$\begin{aligned}
A &= \alpha_s^\top \psi(X) + \alpha_s^{(U)} U + \varepsilon_A \\
Y^a &= \beta_s^\top \phi(X, A = a) + \left(\beta_s^{(U)} + a\beta_s^{(AU)}\right) U + \varepsilon_Y
\end{aligned} \tag{7}$$

where we let $\psi(X) = [1, X]^\top$ and $\phi(X, A) = [1, X, A, AX]^\top$, the noise variables were sampled according to $\varepsilon_A \sim N(0, \frac{1}{8})$ and $\varepsilon_Y \sim N(0, \frac{1}{8})$, and the covariates were sampled according $X \sim N(\mu_X, 1)$ and $U \sim N(\mu_U, 1)$. By default, we set the parameters as $\alpha_s = [\frac{1}{2}, \frac{1}{3}]^\top$, $\beta_s = [\frac{1}{2}, \frac{1}{3}, \frac{1}{2}, \frac{1}{3}]^\top$, $\mu_X = 1$, and $\mu_U = 1$. To impose the presence of an unmeasured confounder, we would set the remaining parameters $(\alpha_s^{(U)}, \beta_s^{(U)}, \beta_s^{(AU)})$ to $\frac{1}{4}$ and otherwise set them to 0. To introduce changes in the parameters among environments, we would select one of the parameter values and overrule the above default values by sampling from a uniformly from the range $[0.1, 3.0]$ for each new environment.

### D.3. Generating Twins semi-synthetic dataset

We use data from twin births in the USA between 1989-1991 (Almond et al., 2005) to construct an multi-environment observational dataset with a known causal structure. The dataset contains 46 covariates related to pregnancy, birth, and parents. As many covariates are highly imbalanced and have low variance, we select a subset of the covariates for generating the semi-synthetic dataset.

As the environment label we used the birth state and as covariates we used the following variables (variable names from the dataset documentation are shown in parenthesis): birth month (birmon), father's age (dfageq), number of live births before twins (dlivord_min), total number of births before twins (dtotord_min), gestation type (gestat10), mom's age (mager8), mom's education (meduc6), mom's place of birth (mplbir), and number of prenatal visits (nprevistq).

The treatment and outcome were generated using the same procedure described in Section D.1, with one key difference: the synthetic covariates were replaced by real-world covariates from the Twins dataset. Prior to generating the treatment and outcome, the covariates were standardized. Each time a semi-synthetic dataset was created using the Twins dataset covariates, five of the chosen covariates were randomly selected as the confounders. A polynomial degree of $p = 2$ was consistently used throughout all experiments.

### D.4. Kernel conditional independence testing with the HGIC approach

When using the original implementation of the hierarchical graph independence constraint (HGIC) approach described in Karlsson & Krijthe (2023) as a baseline in our experiments, we noted that their implementation sometimes would not be properly calibrated (i.e., elevated Type 1 error above $\alpha = 0.05$). For this reason, we introduced some modifications to their

Table 1: Comparison of the new and old HGIC implementation using the data-generating process with polynomial basis functions. The average falsification rate and standard error (in parenthesis) is reported from 250 repetitions.

| Method | No unmeasured confounder | Unmeasured confounder present |
|---|---|---|
| Modified HGIC implementation | 0.04 (.01) | 0.88 (.02) |
| Original HGIC implementation | 0.28 (.03) | 0.80 (.03) |

method that resolved this issue. Note that the modification we describe below were only necessary when using HGIC with the kernel conditional independence test, and not the Pearson conditional independence test used in some other experiments.

The elevated Type 1 errors was caused by that HGIC combines p-values from multiple independence tests using Fisher's method, which employs the test statistic $T = \sum_k \log p_k$ where $p_k$ where are the p-values from the tests (Fisher, 1925). This modification allowed HGIC to use all observations in the multi-environment dataset and was observed to help increase the falsification test's power. However, we noticed in our own simulations that a poorly calibrated conditional independence test can cause the combination of p-values to amplify type I errors. So to address this issue, we made two modifications to the original HGIC implementation.

First, we improved calibration by switching to permutation-based calibration, replacing the original Gamma distribution approximation that is also commonly used for KCIT. Furthermore, we adopted the KCIT implementation from the *causal-learn* Python package (Zheng et al., 2024) which allowed for optimizing the kernel width hyperparameter in the test using Gaussian process regression.

Second, although the above changes improved KCIT's calibration, Fisher's method still sometimes amplified type I errors. To mitigate this, we replaced it with Tippett's method, which uses $T = \min_k p_k$ as the test statistic (Tippett, 1931). Like Fisher's method, Tippett's method emphasizes the smallest p-values (Heard & Rubin-Delanchy, 2018) but we found Tippett's method to be more conservative under the null. Testing on a simple conditional independence scenario confirmed that Tippett's method worked better with KCIT, while retaining the benefits of increased power in combining p-values.

To illustrate the difference between our modified implementation and the original implementation described in Karlsson & Krijthe (2023), we include an experiment using the data-generating process with polynomial basis functions described in Appendix D.1. We used $K = 100$ environments with $N = 50$ samples per environment and $d = 1$ observed confounder, and set the polynomial degree to $p = 2$. Here, both implementations combine 25 p-values. As the results in Table 1 show, our implementation achieved a better Type 1 error while retaining similar power to the original implementation.

### D.5. Additional experiments

We have included three additional experiments in this section to complement the experiments in the main paper.

First, repeating the same setup as in the experiment presented in Section 6.2.1, we include results that confirmed that all methods have controlled Type 1 errors in the well-specified setting. These results are shown in Figure 4.

Secondly, we conducted an ablation study to highlight the importance of using bootstrapping on top of the permutation-based test in our procedure. We repeated the same setup as in Section 6.2.3, but implemented permutation-based testing without bootstrapping. As shown in Figure 5, bootstrapping is crucial for maintaining Type 1 errors below $\alpha = 0.05$, even with an increased sample size.

Lastly, we compared all methods under misspecification across several scenarios in the data-generating process described in Appendix D.1. These scenarios included the presence or absence of an unmeasured confounder, whether transportability holds by sampling the intercept term in $\beta_s$ from $N(0, 1)$ across environments, and whether the underlying data-generating process (DGP) was linear ($p = 1$) or nonlinear ($p = 3$) The results in Table 2 show that misspecification led to elevated Type 1 errors (falsifying without an unmeasured confounder) for all methods. Additionally, transportability violations caused higher Type 1 errors for the transportability test, while our proposed algorithm remained unaffected.

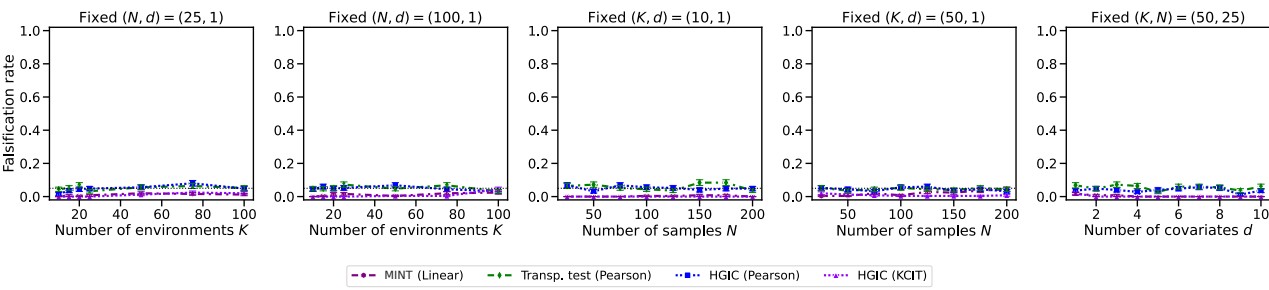

Figure 4: Same experiment as in Figure 1 but with no unmeasured confounding being present. Comparison of falsification rate when varying either the number of environment $K$, the number of samples per environment $N$, or the number of observed covariates $d$. The error bars show the standard error over 250 repetitions.

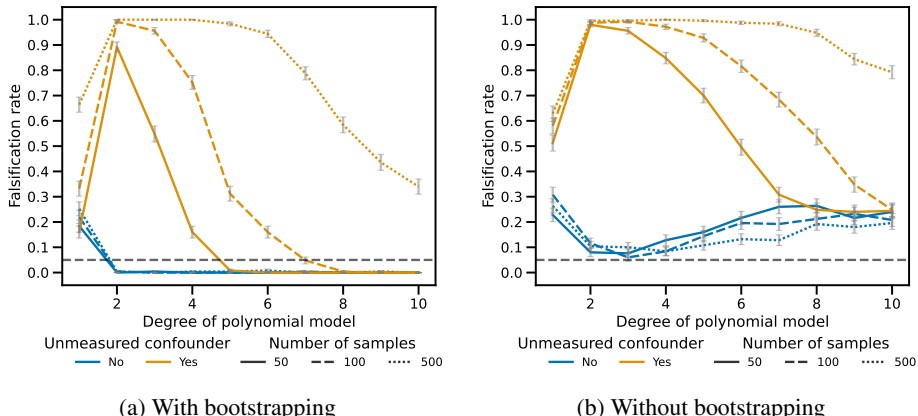

(a) With bootstrapping         (b) Without bootstrapping

Figure 5: An ablation study showing the falsification rate our proposed algorithm using permutation-based testing with bootstrapping versus without bootstrapping. We resample 1000 times when using bootstrapping. The error bars show the standard error over 250 repetitions. The black dotted lines correspond to the chosen significance level $\alpha = 0.05$.

Table 2: Comparison of different approaches under various scenarios with $K = 100$ environments and $N = 100$ samples per environment. The average falsification rate and standard error (in parenthesis) is reported from 250 repetitions.

| | No unmeasured confounder | | | | Unmeasured confounder present | | | |
| Transportability | Holds | | Violated | | Holds | | Violated | |
| DGP | Cubic | Linear | Cubic | Linear | Cubic | Linear | Cubic | Linear |
|---|---|---|---|---|---|---|---|---|
| MINT (Linear) | 0.62 (.03) | 0.01 (.01) | 0.65 (.03) | 0.05 (.01) | 0.60 (.03) | 1.00 (.00) | 0.53 (.03) | 1.00 (.00) |
| MINT (Cubic) | 0.00 (.00) | 0.02 (.01) | 0.04 (.01) | 0.06 (.01) | 1.00 (.00) | 1.00 (.00) | 1.00 (.00) | 1.00 (.00) |
| Transp. test (Pearson) | 0.69 (.03) | 0.04 (.01) | 0.68 (.03) | 0.81 (.02) | 0.65 (.03) | 0.75 (.03) | 0.71 (.03) | 0.86 (.02) |
| Transp. test (KCIT) | 0.05 (.01) | 0.07 (.02) | 0.38 (.03) | 0.43 (.03) | 0.16 (.02) | 0.24 (.03) | 0.34 (.03) | 0.42 (.03) |
| HGIC (Pearson) | 0.66 (.03) | 0.03 (.01) | 0.58 (.03) | 0.04 (.01) | 0.44 (.03) | 1.00 (.00) | 0.43 (.03) | 1.00 (.00) |
| HGIC (KCIT) | 0.02 (.01) | 0.02 (.01) | 0.11 (.02) | 0.02 (.01) | 0.76 (.03) | 0.99 (.01) | 0.35 (.03) | 0.70 (.03) |

