# OpenReview forum: "Falsification of Unconfoundedness by Testing Independence of Causal Mechanisms"
_ICML.cc/2025/Conference — ICML 2025 poster_

### Official Review · Reviewer_GbFw · 2025-03-10

**Overall Recommendation:** 4

**Summary:**

Addressing the common assumption of causal sufficiency, this work proposes an approach to falsify this assumption in causal effect estimation settings. It relies on datasets from multiple environments to test for violations of the Independence of Causal Mechanisms (ICM) principle. The authors study a causal model where treatment and outcome are linear transformations of (known) feature representations and show theoretically that unobserved confounding creates dependencies in certain model parameters. They propose an algorithm to test for unconfoundedness along with a permutation-based calibration scheme to set a test rejection threshold. Besides conceptual improvements over existing approaches, such as avoiding CI testing and having milder assumptions, the method shows promising empirical improvements over existing algorithms in synthetic settings and convincing results on realistic data.

### update after rebuttal ###

I thank the authors for the rebuttal. My assessment remains positive.

**Claims And Evidence:**

All theoretical statements are supported and the proofs are accessible. The authors also made efforts to support their claims empirically, for example illustrating the effect of shifts in different model parameters (Fig. 2).

**Essential References Not Discussed:**

The relevant related work has been cited and put into context.

**Experimental Designs Or Analyses:**

Figs. 1-5. The synthetic data generation is reasonable and there are no apparent issues with the twin dataset setup.

**Methods And Evaluation Criteria:**

The authors evaluate their approach against suitable competitors for falsification of unconfoundedness in a treatment effect setting, using sensible evaluation criteria (falsification rate). The method was also tested in a realistic setting using a twin study proposed in earlier work (Karlsson and Krijthe, 23).

**Other Comments Or Suggestions:**

- ln. 602, left: "then so are the transformations also independent r.v.s"
- ln. 687, left: "where (b) equality follows from that"

**Other Strengths And Weaknesses:**

The problem of addressing untestable assumptions in causal inference, such as sufficiency, is of high significance to the literature and, therefore, a strength of this work. The paper is clearly written, and the presentation of both theory and experiments is well structured. As a potential weakness, the assumptions are rather strong. In particular, the true feature representations $\varphi$ and $\psi$ in the model need to be known. The authors, however, illustrate the implications of this in experiments under misspecification (Fig. 2) and also sketch a kernelized version of their method (Appendix C), looking to remove this assumption in future work.  According to the authors (l. 401 right) the parametric nature also brings certain empirical advantages over existing approaches. Given these points, it seems worthwhile to investigate the given model for the scope of this work.

**Questions For Authors:**

1. Can you comment on how constraining you consider the assumption that the feature representations $\varphi$ and $\psi$ remain the same across environments while only the linear parameters shift? Related to this, how limiting is the fact that your method only works under changes of $\alpha$ and $\mu$ but not $\beta$ (e.g. Fig. 2)?
2. Do you have an explanation for the degrading performance with more covariates (Fig. 1 right) and why this is not an issue for HGIC?
3. The generating process in (3) as described in Appendix D.2. uses quite more samples and environments (1000, 250) than the experiments in D.1. Is such a large number of environments necessary here for the approach to work?

**Relation To Broader Scientific Literature:**

The idea of detecting confounding from violations of the ICM principle has been explored before and the approach is somewhat close to existing work by Karlsson and Krijthe (2023). However, it takes a new perspective with different algorithmic ideas, adding sufficient originality. Overall, there is only a small literature addressing untestable assumptions in causal inference and discovery, and the paper adds useful new insights.

**Theoretical Claims:**

Thm. 4.2, 4.3, 4.4, where I did not find any issues.

---

> ### Author Rebuttal · Authors · 2025-03-29
>
> We thank the reviewer for their invaluable feedback, suggestions for improvement and questions. Below are answers to your questions.
>
> # 1: Fixed feature representations with linear parameter shifts
> We agree that it is important to consider what it entails to assume a model class with fixed feature representation across environments with linear parameters shifts. Our key idea -- testing independence of causal mechanisms -- benefits from the fixed feature representation, as it enables efficient testing by focusing solely on the linear parameters. However, a fixed feature representation with linear parameter shifts is not fundamental to our proposed falsification strategy for testing mechanism independence. As explored in  Appendix C, one could, in principle, replace the fixed feature representation with a potentially infinite-dimensional implicit representation using kernel methods.
>
> That said, we do not consider the assumption of a fixed feature representation to be particularly restrictive. To illustrate this, suppose a feature $X_1$ is highly predictive of treatment assignment in environment A but not in environment B. The fixed representation for environment A and B can still accommodate this by including $X_1$ (and its nonlinear transformations), and then only the linear parameter in environment A will utilize this feature whereas the linear parameter in environment B will not. This suggests having a fixed representation is not overly restrictive, as long as it is “rich enough” to accommodate different environments. In practice, of course, the richness of the representation will affect the efficiency of the test, making it a trade-off between expressiveness and statistical power. Hence, methods to learn a common representation or implicit representations are interesting future research directions.
>
> # 1: Unable to falsify under changes of $\beta$
> In an ideal world, we would hope that our method could also falsify under solely changes on $\beta$. However, based on our current understanding, this may represent a fundamental limitation of this type of falsification approach, as both our findings and those of Karlsson and Krijthe (2023) suggest the same conclusion. But from a practical view-point, a key characteristic of observational studies is that treatment assignment is non-randomized. In our setting, where we look at multiple observational studies, it would seem plausible that each study has its own unique treatment assignment mechanism. Thus, in practical use cases one could expect $\alpha$ (if not $\mu$) to likely vary across environment.
>
> # 2: Degradation with more covariates
> The decline in falsification rate as the number of covariates increases in the right-most plot of Figure 1 can be explained as follows: As the number of covariates grows, estimating the linear parameters $(\omega,\gamma)$ becomes more challenging, leading to greater sample variance in the estimates. This, in turn, increases the uncertainty in estimating the covariance matrix $\text{Cov}(\omega,\gamma)$  which is crucial for computing our test statistic. Consequently, the test loses power as the number of covariates rises.
>
> For HGIC, the choice of conditional independence test significantly impacts its response to increasing covariates. For example, we found that using HGIC with the kernel conditional independence test (KCIT) performed poorly as the number of covariates increased; an expected outcome since large conditioning sets are known to harm KCIT (see Zhang et al., 2011). Meanwhile, HGIC remained effective when using the Pearson partial correlation test, even as the number of covariates grew. But this robustness was not unique to HGIC: the transportability test with Pearson’s test also remained unaffected by additional covariates. Thus, we could only speculate that the test statistic used in the Pearson test is very well-suited for the linear synthetic data we generated.
>
> # 3: Large sample sizes in data-generating process in Appendix D.2
> In the experiment shown in Figure 2a, where we used data as described in Appendix D.2, our goal was to explore the theoretical limits of our approach in light of the results from Section 4.3. Specifically, the identifiability guarantees for falsifying unmeasured confounding under various linear parameter shifts. To minimize finite-sample effects, we increased the sample size and the number of environments. However, due to computational constraints, we could not scale them indefinitely. Notably, this was the only experiment where we took this approach. In our other experiments, our method remains effective even with much smaller sample sizes and fewer environments. For instance, in the left-most plot of Figure 1, our approach was able to successfully falsify unconfoundedness with just 10 environments and 25 samples per environment.
>
> # References
> Zhang, Kun, et al. "Kernel-based conditional independence test and application in causal discovery." UAI, 2011.

---

### Official Review · Reviewer_33m1 · 2025-03-14

**Overall Recommendation:** 3

**Summary:**

The paper proposes a test for unconfoundedness based on the usual assumptions of causality from the potential outcomes perspective, but more critically assumptions about the independence of causal mechanisms and a specific functional form (including specifying functional form well). The test requires observing more than one environment. The test statistic is based on a Frobenius norm of the covariance matrix of the parameters of the causal mechanisms. The test statistic is calibrated through permutation. Finally, the authors investigate the falsification rate of the method and baselines with synthetic data, modifying the number of environments, number of observations per environment, number of covariates, and violations to the specification of the model. They also test it on a semi-synthetic dataset.

**Claims And Evidence:**

In general the claims are supported by evidence, both theoretically and empirically. Maybe a minor complaint about claims vs. evidence is the fact that the authors claim they tried their method in real-world data whereas the closest they get to real world data is semi-synthetic data.

**Essential References Not Discussed:**

The authors cite several related research. I would personally add a couple more references, for example in page 2, column 2, line 97 when they talk about multi-environment data and looking only at the relevant portion for causal effect estimation, that’s precisely the idea of Invariant Causal Prediction (ICP) in Peters et.al. (2016) (from JRSSSB) which does the same task and were one of the first to explicitly include environment information for causal tasks. Other research that could have been included is the more recent work on falsifiability of causal discovery algorithms like Faller et.al. (2024) in AISTATS.

**Ethical Review Concerns:**

No concerns

**Experimental Designs Or Analyses:**

The experimental design and analysis seem very reasonable to me. Both the test statistic and the permutation based calibration make sense. Would be interesting to know whether the authors tried other test statistics or whether they think that other tests would work and potentially would be more efficient in some way. Additionally, testing for the number of environments, sample size, number of covariates and misspecification is something I would expect to see.

**Methods And Evaluation Criteria:**

Yes, the methods and the evaluation criteria make sense for the proposed problem.

**Other Comments Or Suggestions:**

None.

**Other Strengths And Weaknesses:**

Strengths:
- The theoretical ideas of the paper are interesting and seem reasonable to me.
- The empirical tests make sense with respect to their claims.

Weaknesses:
- I think the empirical evaluation reveals what I consider to be one of the biggest weaknesses of the paper. Particularly how data hungry the method is. I can imagine that because of the way the test statistic is designed –and the permutation based calibration– one needs to have several environments and several observations per environment to get a reasonable falsification rate.

**Questions For Authors:**

The results of section 6.2.2. are really difficult to interpret. What I understand is that you change the parameters as changes in the mechanisms to see whether you can detect unconfoundedness using your test. But why do we expect changes in the alphas to give us falsification while changes in betas don’t?

**Relation To Broader Scientific Literature:**

I think the ideas presented in the paper are interesting. In the end, confounding is arguably one of the biggest problems in causality, and proposing a method for detecting confounding based on heterogeneous data is a valuable addition to the research on causality.

**Theoretical Claims:**

I didn’t check the proofs of any of the theoretical results, but they all seem natural to me (of course with the exception of Lemma 4.3., which requires a lot of computation.

---

> ### Author Rebuttal · Authors · 2025-03-29
>
> We thank the reviewer for the response, in particular their excitement for our proposal for extending our method to implicit feature representations. Whereas it felt as to be outside the scope of the current manuscript, we agree this is one of the more promising directions of our work and we also hope to encourage others to see this possibility. Below you can find answers to your comments. In addition, based on your feedback, we are doing the following changes to our camera-ready version:
> - we state more explicitly in the abstract and experimental section that we use semi-synthetic data.
> - we add the references you suggested to the related works section.
>
> # Clarification on results in section 6.2.2
> The primary objective of the experiment described in Section 6.2.2 is to empirically validate the theoretical results presented in Section 4.3. To achieve this, we simulate data based on the model specified in Equation (3). The different parameters on the x-axis in Figure 2a represent which of the parameters in Equation (3) are allowed to vary across different environments. This is done under both the absence and presence of unmeasured confounding.
>
> Theorem 4.2 predicts that any changes associated with the $\alpha$ parameters should enable the detection of an unmeasured confounder, if one is present. Our experimental results confirm this prediction. The underlying reason is that, unlike the $\beta$ parameters, the $\alpha$ parameters appear in both the treatment and outcome mechanisms, as established in Lemma 4.3.
>
> # Using other test statistics
> There are indeed alternative test statistics for testing the independence between two random variables: the main requirement is that the test statistic works with multivariate random variables (due to $\omega$ and $\gamma$ being multivariate). Our reasoning for using the Frobenius norm of the covariance matrix as a test statistic stems from the fact that it only considers linear dependencies, which are typically the easiest to identify. One could also consider other test statistics that consider nonlinear dependencies, but this extra flexibility could lead to the method being more data hungry. For instance, we also considered the Hilbert-Schmidt Independence Criterion (HSIC) which allows for testing nonlinear dependencies using an appropriate kernel (when using a linear kernel one would obtain the test statistic which we already use). However, some initial experiments showed that HSIC was also not particularly robust. Another alternative test statistic that we did not explore but could be considered is distance correlation, which can also detect both linear and nonlinear dependencies.
>
> # Connection with ICP
> We agree that Invariant Causal Prediction (ICP), as introduced by Peters et al. (2016), should be acknowledged for its role in highlighting the premise of utilizing data from multiple environments. We have thus added a reference to their paper in the related works. While both ICP and our approach consider multiple environments, they are based on fundamentally different principles. Our method explicitly exploits changes across environments, whereas ICP is designed to identify what remains invariant.
>
> To further illustrate this contrast, Peters et al. (2016) state on page 2:
>
> *“We exploit, in other words, that the conditional distribution of the target variable of interest (often also termed “response variable”), given the complete set of corresponding direct causal predictors, has to remain identical under interventions on variables other than the target variable”*
>
> This stands in stark contrast to our framework. Unlike ICP, which believes the conditional distribution of the target variable to remain unchanged across environments, we explicitly state in Assumption 4.1 that the conditional distributions of the target variables (in this case outcome and treatment variables) can vary across environments rather than remain invariant.
>
> # Comment on our method being “data hungry”
> While improving sample efficiency is an important goal for future work, we emphasize that our method already outperforms the baselines across different numbers of environments and samples per environment. Even in the most challenging small-sample case we tested – 10 environments with 25 samples each – our method achieved a falsification rate of 0.8, compared to 0.6 and 0.4 for the baselines. Having said that, access to multiple environments is critical for any falsification method based on variation between environments to work, though the number of required environments will likely depend on how much variation there is between the environments.

---

### Official Review · Reviewer_sWrJ · 2025-03-21

**Overall Recommendation:** 4

**Summary:**

This paper addresses the problem of unmeasured confounding in observational data.  Most causal estimation methods assume that there are no unmeasured confounders, an assumption that is hard to test in practice.  The authors propose a method for testing this assumption in situations where data from multiple environments is present.  Previous work in this area relied on unreliable conditional independence testing, so this work proposes a novel approach based on hypothesis testing of statistical covariance instead.  If the model is well-specified, the authors show that this approach consistently outperforms other approaches.

I appreciate the authors' comments.  The other reviewers seemed to share similar concerns around the clarity of the connection/difference from Karlsson & Krijthe 2023, and it seems like the authors are aware of the need to explain this in more detail.  With that and an explicit discussion of assumptions are feature representation, I'm happy to raise my score.

**Claims And Evidence:**

The claims made by this paper appear well supported.  The problem they're seeking to address is well-motivated and defined, and the approach seems sound.

**Essential References Not Discussed:**

I'm not sure how essential this is, but in the Related Work, I was surprised that there was no discussion of identifiability, since that seems like a closer analogue to falsification than sensitivity analysis.

**Experimental Designs Or Analyses:**

The baselines used for the experiments make sense, and the experimental design overall is sound.  I do wish the authors had done more experimentation with model-misspecification, since the need for a correctly-specified model seems like a major weakness of this approach, so more robust testing of the consequences of model-misspecification would help reduce those concerns.

**Methods And Evaluation Criteria:**

The synthetic data used for the experiments appears reasonable.  I do think any detail at all about the synthetic data used should be included in the main body of the paper and not relegated to Appendix D.1, but it appears like a reasonable choice for synthetic data.  The twins data set is a nice choice for an empirical evaluation.  I wish more than 2 other methods were available for comparison, but given the specific nature of the problem being addressed, it's understandable.

**Other Comments Or Suggestions:**

You should name your algorithm so it's not just called "Ours".  If it's largely an extension to HGIC, naming it HGIC-XXX (for some relevant XXX acronym) would be fine, but it's clunky to not actually have a name in the paper.

**Other Strengths And Weaknesses:**

Overall, I think this is a solid paper.  The problem is well-motivated, the paper layout and descriptions are very clear, and the approach seems reasonable and effective.

I wish there were more discussion around the feature representations psi and phi.  The need for correctly-specified feature representations isn't really discussed in the paper until the experimental results in 6.2.3 and the post-experiment discussion.  While the narrative of the paper as a whole flows well, Section 4.1 seems to be missing at least a sentence or two of lead-in.  An assumption of this approach seems to be that we have access to these feature representations, but it's not actually directly called out as an assumption anywhere that I can see.  As you mention in the discussion, we could learn these from data, so it's not a fatal flaw.  But some discussion in Section 4 about how we get psi and phi, and experimental results showing that it's at least somewhat possible with empirically learned psi and phi, would help a lot.

I think some clarity around what assumptions are made by your algorithm, and which assumptions are ones you'll be attempting to falsify, would help.  Assumption 3.1 is a common assumption and one you'll falsify, but that's not directly stated in Section 3.2 until the final sentence, making it a bit unclear.  Similarly, Assumption 4.1 is another one that your algorithm attempts to falsify, but just a little bit before that in the body of the text (lines 170-171), you state another assumption ("the feature representations psi(X) and phi(X,A) are considered to be fixed across environments"), but I think this assumption is one you're /actually/ making, not one you're falsifying.

The first time psi-~ and phi-~ appear, I believe, is lines 210-211 in Section 4.2, but you don't actually define them at this point (you just say that we assume we have access to them).  In the next paragraph, you discuss them more and it seems like they are estimates of phi and psi, but actually defining them before you use them would help.

Are you using the wrong y-axis label in Figure 2?  Figures 1 and 2 both have the y-axis labeled "Falsification rate", but in Figure 1, higher is clearly better (which makes sense since falsification is the goal), while in Figure 2, lower is better.  Figure 2 seems to actually be the p-value from the hypothesis test? (hence why you show the alpha=0.05 significance level)

**Questions For Authors:**

My first two are primarily what I went over under "Relation to Broader Scientific Literature:

1. How does your proposed algorithm differ from HGIC in Karlsson & Krijthe 2023?
2. In what ways are our first two listed contributions different from the similar formalisms/proofs in Karlsson & Krijthe 2023?

3. Could you discuss the transportability assumption, and how it relates to your work, a bit more?  In Related Work, you contrast your work with prior work in transportability and point out the key difference as being that your approach "assumes independence of causal mechanisms" while this work "require[s] transportable treatment effect or access to randomized data."  In what ways is the 'independence causal mechanisms' assumption more reasonable/achievable in practice than the 'transportable treatment effect' assumption?

4. Unless I'm misunderstand what's being shown in Figure 2 (see my confusion about the y-axis label for Figure 1 vs Figure 2), it appears as though, in Figure 2b, in the presence of unmeasured confounding, the falsification rate is actually significantly worse as the number of samples increases.  For example, for a 6-degree polynomial, only using 50 samples has a falsification rate (p-value??) of nearly 0 (well below the 0.05 line) while 500 samples is over .9!  Why is that?

**Relation To Broader Scientific Literature:**

This is probably my biggest open question.  From the related work, it seems like your approach is quite similar, at least in problem setup and high-level approach, to Karlsson & Krijthe 2023.  The related work, in fact, makes it seem almost incremental (i.e., same exactly approach but they did CI tests and you did a different sort of test), but I'm not sure if that's actually the case.  It would be helpful to more clearly describe how this paper extends, and differs from, Karlsson & Krijthe 2023.  For example, your first two listed contributions are: "we formalize a Neyman-Rubin causal model for multi-environment data under the principle of independence causal mechanisms" and "we prove that the presence of unmeasured confounding has testable implications in the form of dependencies between the model's observed parameters".  Skimming through Karlsson & Krijthe 2023, both of those already seem to be covered by that work.  Is there some subtle difference between their version and this one that would qualify both of these as novel contributions?

**Theoretical Claims:**

While I did not dig into the proofs in the supplementary material, the theoretical claims in the main paper appear correct

---

> ### Author Rebuttal · Authors · 2025-03-29
>
> We thank the reviewer for the invaluable feedback and questions. We have provided answers and proposed changes related to your questions below. In addition, we also do the following changes for the camera-ready version:
> - update Section 4 with more explicit assumptions on the feature representations
> - name our method to MINT (Mechanism INdependence Test)
> - add references on identifiability in related works section, specifically on partial identification under unmeasured confounding
>
> # 1, 2: Differences in our work and Karlsson and Krijthe (2023)
> While we adopt the same setting as [KK23], a key conceptual difference in our approach makes our falsification strategy more sample-efficient. The framework of [KK23], which is based on DAGs, leads to a constraint-based causal discovery approach that relies on faithfulness and the causal Markov property. Importantly, they test independent causal mechanisms *indirectly* via a conditional independence statement involving treatment, outcome, and covariates in their assumed DAG. In contrast, one of our main contributions is the insight that it is possible to *directly* test the independence of causal mechanisms by inspecting the parameters of the treatment mechanism $E[A\mid X,S=s]$ and outcome mechanism $E[Y^a\mid X,S=s]$, which allows us to leverage functional assumptions. Here we prove the same theoretical guarantees for falsification remain when going from the indirect testing to the direct testing of ICM.
>
> This new conceptual shift results in a substantially different algorithm. While [KK23] performs conditional independence tests on the data, our algorithm uses a two-stage approach: first estimating nuisance models (parameters of the treatment and outcome mechanism) before applying an unconditional independence test. Our approach appears to be more efficient, provided these models are well specified, even in comparison to HGIC when a well-specified conditional independence test was used.
>
> There are two potential reasons for this efficiency. First, an unconditional independence test is statistically easier than a conditional one. Second, and more subtly, HGIC does not naturally allow for the efficient use of all available data in each environment because the conditional independence test in HGIC can only use 2 samples per environment, so it aggregates results from multiple tests to utilize all samples in each environment. Our algorithm avoids this issue entirely, as it leverages all data within a single independence test.
>
> Based on this, we will make the following changes to the manuscript:
> - add a paragraph in Section 4 highlighting the above-mentioned differences between our work and [KK23].
> - reformulate first part of our contributions to: *“by formalizing the problem using a Neyman-Rubin causal model for multi-environment data, we show that falsification of unconfoundedness is possible by testing dependencies between causal mechanisms directly, rather than indirectly, by combining the principle of independent causal mechanism with functional assumptions on the mechanisms.”*
>
> # 3: Transportability versus ICM
> To argue why ICM can be a more reasonable assumption than transportability, we will illustrate why we think that ICM is typically a weaker assumption than transportability. We will add this explanation to our paper when introducing the ICM assumption. We also want to point the reviewer to Appendix A in the supplementary materials where we discussed transportability in more depth.
>
> Specifically, transportability violations occur (according to most definitions as we mention in Appendix A) when the outcome mechanism $E[Y^a\mid X, S=s]$ varies across environments. A reason why this can happen is due to the presence of unmeasured effect modifiers, which often is a plausible scenario in many real-world settings. Very importantly, our Assumption 4.1 on ICM specifically allows $E[Y^a\mid X, S=s]$ to change across environments. Thus, we can establish that even when transportability is violated, it is clear that ICM may still hold. What this means in practice is that our method is insensitive to whether transportability holds or not, whereas transportability-based tests will exhibit false positives (i.e. falsify unconfoundedness in the absence of unmeasured confounding) under violations of transportability.
>
> # 4: Clarification on Figure 2
> Thanks to your comment, we understand the confusion. The falsification rate is the probability that a method will falsify unconfoundedness, regardless of whether this is correctly (there is an unmeasured confounder) or incorrectly done (there is no unmeasured confounder). It should have been more clear that under the absence of unmeasured confounding, we want a falsification rate below the significance level $\alpha$. Under the presence of unmeasured confounding, a higher falsification rate is better.
>
> Based on this, we will update the figure caption of Figure 2b to clarify how to interpret the falsification rate on the y-axis.

---

### Official Review · Reviewer_mtnu · 2025-03-25

**Overall Recommendation:** 3

**Summary:**

This manuscript presents an algorithm for falsifying the assumption of no unmeasured confounding in a setting of observational data from multiple environments. To this end, the authors employ the Rubin potential outcome causal model, assume positivity, consistency and no unmeasured confounding; subsequently they model the functional relationships between (a) the confounders and the treatment and (b) treatment, confounders and the outcome. Finally, they introduce the key assumption that the parameters associated with functions (a) and (b) are independent from each other for all environments. The main theoretical result of the paper follows: given all the aforementioned assumptions, the learned parameters associated with (a) and (b) are independent. Then a Gaussian linear model is assumed which makes it possible to equate the dependence of the learned parameters with unobserved confounding. The paper concludes with a series of experiments (3 on synthetic data and one on real data) comparing the proposed approach in terms of efficiency (falsification rates) to two benchmarks.



##### After rebuttal #####


I would like to thank the authors for their rebuttal.


I acknowledge the algorithmic contribution of the paper. The authors' response has shed a little more light on the difference to the theoretical results in [KK23], but a closer examination and establishing a formal relationship between them (showing whether the difference only stems from switching paradigms from causal DAGs to potential outcomes) could improve the paper. All in all, I am willing to raise my rating by a notch.


My current understanding is that there is no clear relationship between the theoretical results (the algorithmic ones differ of course) in [KK23] and this submission, but I still have a feeling that they are related and the relationship could be formalized by specifying e.g. additional conditions for some sort of equivalence.

The lack of conditional independence testing is a clear difference, and maybe the biggest contribution of the manuscript.

I would like to clarify that I am not in any way associated with the authors of (Wang, Blei, 2018) or of any other follow-up paper I mentioned, nor have ever I published in the post 2018 “deconfounder” line of research. I simply thought the high level intuition “if the model (or causal mechanisms) do not factorize perfectly, this might be because of unobserved confounding” is something the current submission and (Wang, Blei, 2018) have in common and was wondering whether the authors had any thoughts on this. I think that citing any of the works only makes sense if accompanied by a detailed discussion.

I acknowledge the algorithmic contribution of the paper. The theoretical discussion has shed a little more light on the difference to [KK23], but a closer examination of the results could improve the paper. All in all, I am willing to raise my rating by a notch.

**Claims And Evidence:**

Yes.

**Essential References Not Discussed:**

The paper’s theoretical results are very similar to (Karlsson, Krijthe, 2023), and while it is cited, I feel that a more thorough discussion of the overlap (especially concerning theory) would improve the manuscripts’ clarity.

**Experimental Designs Or Analyses:**

The experimental design presented in the main body of the paper seems OK (Chapter 6; two synthetic experiments and a real-data one). I did not check the details in Appendix D.

**Methods And Evaluation Criteria:**

Yes.

**Other Comments Or Suggestions:**

Typos:


l. 112 refereed -> referred


l. 162 accounted -> accounted for


l. 324 refereed -> referred

**Other Strengths And Weaknesses:**

Strengths


The paper tackles an important problem of testing the existence of unmeasured confounding.
The main contribution of the paper is a performance improvement over a similar method (Karlsson, Krijthe, 2023) on synthetic and real data.



Weaknesses


The main weakness of the paper is its limited novelty. The theoretical results mirror those of (Karlsson, Krijthe, 2023) (Theorem 1 and Theorem 2). The main difference is that the current paper adopts the potential outcome framework instead of a Pearlian graphical model (for Thm 4.2) and a Gaussian linear model (Lemma 4.3, Thm 4.4).


The paper would be easier to follow if a discussion of (Karlsson, Krijthe, 2023) were held along with enunciating the current paper’s relation to it in terms of theory. It would also improve clarity if the paper’s focus on experimental improvement over (Karlsson, Krijthe, 2023) was stated.


I feel that the role of environments is not discussed clearly enough. It seems to me that the proposed method beats (Karlsson, Krijthe, 2023) more clearly for a small number of environments (Figure 1). Do the environments facilitate reasoning about statistical efficiency?


It would be interesting to compare the paper with another approach for unmeasured confounding modelling, namely (Wang, Blei, The Blessings of Multiple Causes, 2018). The paper caused a lot of response leading to a number of limiting results for unmeasured confounding modelling (D’Amour 2018 Comment: Reflections on the Deconfounder; D'Amour 2019 On Multi-Cause Causal Inference with Unobserved Confounding:
Counterexamples, Impossibility, and Alternatives; Ogburn et al. 2020 Counterexamples to "The Blessings of Multiple Causes" by
Wang and Blei). The underlying intuition of (Wang, Blei, 2018) is similar to the one presented here: if the model for the confounded treatment-response factorizes into independent parts, there has to be (multiple) unmeasured confounding. Do any of the model limitations apply here? Do the environments play a role?

**Questions For Authors:**

1. Given the equivalence of the Pearl model and the potential outcome model (Pearl 2009), does Theorem 4.2 follow from Theorem 1 in (Karlsson, Krijthe, 2023)? Under the assumption of a Gaussian linear model, do Lemma 4.3 and Thm 4.4 follow from Theorems 1 and 2 in?


2. What is the relation of the unmeasured confounding falsification to (Wang, Blei, 2018) and the following discussion?

**Relation To Broader Scientific Literature:**

The paper’s theoretical results are very similar to (Karlsson, Krijthe, 2023), and while it is cited, I feel that a more thorough discussion of the overlap (especially concerning theory) would improve the manuscripts’ clarity.


A review of de-confounding, (e.g. Louizos et al. 2017), dealing with ignorability or propensity scores is missing, but this is not fatal and can be explained with limited space.

**Theoretical Claims:**

I read the three proofs in Appendix B (Theorem 4.2, Lemma 4.3, Theorem 4.4). I did not check the math details in the proof of Lemma 4.3, but it seems straightforward (multiplying two normal densities and plugging in to the assumed model). I also skipped the proof of Lemma B.2.


The proofs appear to be OK.

---

> ### Author Rebuttal · Authors · 2025-03-29
>
> We thank the reviewer for the valuable feedback and for spotting a number of typos which now have been addressed. We have provided answers and proposed changes related to your questions below.
>
> # 1: Connection between our theory and that of of Karlsson and Krijthe (2023)
> While the possibility of testing for unconfoundedness may be less surprising in light of [KK23], it is reassuring that we obtain similar identifiability guarantees under different assumptions and proof techniques. Our problem formulation not only leads to a novel test that directly targets the dependence between mechanisms but also introduces different conditions under which the results hold. In this context, we would kindly ask you to also read our reply to reviewer sWrJ, where we emphasize a key conceptual difference between our approach and that of [KK23]. Specifically, our framework *directly* tests the independence of causal mechanisms (ICM), necessitating a new theoretical treatment of the problem.
>
> We now go over two key differences in the assumptions between our theory and [KK23] to clarify why our findings do not trivially follow from [KK23], even under a linear-Gaussian model:
> 1. Most importantly, there is a difference in the formulation of the ICM assumption. [KK23] assumes that, in the absence of unmeasured confounding, variations in the conditional distributions $P(A\mid X,S=s)$, $P(Y\mid X,A,S=s)$ and $P(X\mid S=s)$ are independent of each other. In contrast, our formulation in Assumption 4.1 only requires changes in $P(A\mid X, S=s)$ and $P(Y^a\mid X, S=s)$ to be unrelated, without making any assumptions about $P(X\mid S=s)$. As [KK23] highlights in their Section 4.1 ("Influence of assumptions"), this additional condition on $P(X\mid S=s)$ is necessary for their method to work. Meanwhile, our approach achieves the same falsification guarantees without requiring this additional condition because the results in Lemma 4.3 holds for any $P(X\mid S=s)$.
> 2. Also, [KK23] uses a constraint-based causal discovery approach relying on assuming the faithfulness and causal Markov property. In contrast, by instead leveraging functional assumptions, our approach does not require these assumptions.
>
> Based on the above discussion, and in addition to the changes mentioned in our response to reviewer sWrJ to clarify our contributions relative to [KK23], we will add a paragraph in Section 4 of our manuscript highlighting the key differences in assumptions between [KK23] and our work.
>
> # 2: Relationship to de-confounding literature
> We thank the reviewer for highlighting the work of Wang and Blei (2018) and the surrounding discussion on it. As suggested, we will add references to these papers in our manuscript.
>
> We are aware of this line of work, but we think the setting and motivation considered by these papers do not reflect our work. First, the argument used in Wang and Blei does not apply to our setting: we do not have multiple observed causes, but rather multiple environments. Second, they focus on identifiability about the interventional distribution under unmeasured confounding. In contrast, our goal is slightly less ambitious, we just want to determine whether there is an unmeasured confounder in the first place.
>
> That said, it would be interesting to explore if there is a similarity between the factorization exploited by Wang and Blei and the ICM assumption, which also assumes a type of factorization. This could help us understand if there are implications for identifying interventional distributions in our setting. Here, the key question becomes what the access to data from multiple environments will add and whether that will allow for identifiability.
>
> # Do the environments facilitate reasoning about statistical efficiency?
> The number of environments plays a crucial role in the success of falsification. As a thought experiment, consider a scenario where $(\omega_s,\gamma_s)$ are directly observed, allowing us to bypass estimating them in the first stage of our algorithm. In this case, the number of environments effectively determines the sample size available for testing independence between these two random variables.
>
> If $\omega_s$ and $\gamma_s$ are only weakly dependent – such as when unmeasured confounding is weak – detecting their dependence would require a large number of environments. Conversely, if the confounding is strong, fewer environments may suffice to reveal dependence in these model parameters.
>
> Once we add back the fact that we need to estimate $\omega_s$ and $\gamma_s$, this introduces additional sampling variance into the independence test. As a result, with fewer samples per environment, detecting a dependence becomes more challenging, and a larger number of environments will likely be required to achieve a high statistical power of our test. This is also what we observe in our experiments shown in Figure 1: decreasing number of environments or the sample size leads to lower falsification rate.

---

> > ### Comment · Reviewer_mtnu · 2025-04-09
> >
> > I would like to thank the authors for their rebuttal.
> >
> >
> > Thank you for pointing out that lack of the reliance on $P(X|S=s)$ is another difference wrt [KK23]. My current understanding is that there is no clear relationship between the theoretical results (the algorithmic ones differ of course) in [KK23] and this submission, but I still have a feeling that they are related and the relationship could be formalized by specifying e.g. additional conditions for some sort of equivalence.
> >
> >
> > The lack of conditional independence testing is a clear difference, and maybe the biggest contribution of the manuscript.
> >
> >
> > I would like to clarify that I am not in any way associated with the authors of (Wang, Blei, 2018) or of any other follow-up paper I mentioned, nor have ever I published in the post 2018 “deconfounder” line of research. I simply thought the high level intuition “if the model (or causal mechanisms) do not factorize perfectly, this might be because of unobserved confounding” is something the current submission and (Wang, Blei, 2018) have in common and was wondering whether the authors had any thoughts on this. I think that citing any of the works only makes sense if accompanied by a detailed discussion.
> >
> >
> > I acknowledge the algorithmic contribution of the paper. The theoretical discussion has shed a little more light on the difference to [KK23], but a closer examination of the results could improve the paper. All in all, I am willing to raise my rating by a notch.

---

### Decision · Program_Chairs · 2025-05-01

**Decision:**

Accept (poster)

**Comment:**

The authors propose a simple method - leveraging multiple observational data sets to test independence of the coefficient parameters that are assumed to be independently drawn under the null (ICM assumption) - that rejects the null hypothesis only if one of their assumptions (unconfoundedness, standard assumptions such as consistency, positivity, independence of mechanism parameters, or well-specification of the treatment effect and outcome models) do not hold. That means that, if the hypothesis is rejected, it may be due to unmeasured confounding - although it is not possible to know for sure that confounding was the "culprit" or one of the other many possible reasons, it is still valuable information to know what not to do with the data for treatment effect estimation.

Apart from the reviewer comments you have already addressed, from a quick reading, it seems to me that a very strong assumption that may jeopardize the usefulness of the method in practice is that the null may either be rejected if the feature representations $\tilde{\phi}, \tilde{\psi}$ are not correct (i.e. misspecification of A|... and Y|...) or the model not expressive enough unless $\alpha, \beta$ are made very large - in which case testing independence becomes harder and that power suffers (even in the case of dependence, it is not spottable when d>>n). This is implicitly suggested in Figure 2b) - it would be nice to plot "power" as well in that case (I couldn't find a plot). Hence I wonder about the usefulness of this method for real-world data (i.e. when the polynomial model is not of degree 2 but known to be quite complicated - and hence enforcing the dimensionality of $\alpha,\beta$ to be large. In this case, how does this new method compare with HGIC/Pearson tests? This seems to be reflected in the last plot in Figure 1. I wish the authors could provide a better power analysis in (semi-)synthetic data (beyond just table 1) and a more experiments and discussion of the effectiveness of the new test on real-world data.